# Systematic characterization of pan-cancer mutation clusters

Marija Buljan[1,2] (iD), Peter Blattmann[1] (iD), Ruedi Aebersold[1,3,*] (iD) & Michael Boutros[2,4,5,**] (iD)

## Abstract

Cancer genome sequencing has shown that driver genes can often be distinguished not only by the elevated mutation frequency but also by specific nucleotide positions that accumulate changes at a high rate. However, properties associated with a residue's potential to drive tumorigenesis when mutated have not yet been systematically investigated. Here, using a novel methodological approach, we identify and characterize a compendium of 180 hotspot residues within 160 human proteins which occur with a significant frequency and are likely to have functionally relevant impact. We find that such mutations (i) are more prominent in proteins that can exist in the on and off state, (ii) reflect the identity of a tumor of origin, and (iii) often localize within interfaces which mediate interactions with other proteins or ligands. Following, we further examine structural data for human protein complexes and identify a number of additional protein interfaces that accumulate cancer mutations at a high rate. Jointly, these analyses suggest that disruption and dysregulation of protein interactions can be instrumental in switching functions of cancer proteins and activating downstream changes.

**Keywords** cancer genomics; hotspot analysis; interface mutations; protein complexes

**Subject Categories** Cancer; Computational Biology; Genome-Scale & Integrative Biology

**Mol Syst Biol (2018) 14: e7974**

## Introduction

Cancer genome sequencing has had a major role in mapping cellular pathways leading to tumorigenesis (Lawrence *et al*, 2014; Leiserson *et al*, 2015) and in elucidating diverse molecular mechanisms that can drive oncogenic transformation (Alexandrov *et al*, 2013) and drug resistance (Garraway & Lander, 2013). These mechanisms include genomic rearrangements, smaller insertions and deletions, or single point mutations. Eventually, they all lead to changes in the expression levels or to altered functions of cancer driver genes and their products. Analysis of different cancer genomics datasets has further underscored a high degree of heterogeneity in the mutation frequency and spectrum among different cancer types (Garraway & Lander, 2013; Lawrence *et al*, 2013) and uncovered a long tail of low-frequency driver mutations (Garraway & Lander, 2013). As a corollary, in spite of the great progress in charting mutational events that define different cancer types, the task to distinguish driver and passenger mutations in an individual genome remains a formidable challenge. Furthermore, even when a high mutation rate across a number of patients indicates the gene is a cancer driver, functional impact of individual mutations and their connection to the affected cellular pathways are not readily evident (Garraway & Lander, 2013; Cancer Genome Atlas Research, 2014b; Alvarez *et al*, 2016).

Large-scale cancer genome initiatives, specifically The Cancer Genome Atlas (TCGA, https://cancergenome.nih.gov/) and International Cancer Genome Consortium (International Cancer Genome *et al*, 2010; ICGC, http://icgc.org/), have increased statistical power in the analyses of cancer mutations and have driven the development of innovative approaches for the study of patient data (Dees *et al*, 2012; Hofree *et al*, 2013; Kandoth *et al*, 2013; Lawrence *et al*, 2013, 2014; Chen *et al*, 2014; Sanchez-Garcia *et al*, 2014). In particular, a number of recent methods address the fact that even within a specific driver gene not all mutations will have an equal effect (Kan *et al*, 2010; Burke *et al*, 2012; Porta-Pardo *et al*, 2017). To account for this, they implement positional mutation biases as criteria for the detection of candidate driver genes (Davoli *et al*, 2013; Tamborero *et al*, 2013; Kamburov *et al*, 2015; Tokheim *et al*, 2016; Gao *et al*, 2017). Some of these approaches are agnostic to prior knowledge (Araya *et al*, 2016), while others focus on regions of known functional relevance, such as protein domains (Miller *et al*, 2015; Yang *et al*, 2015), phosphosites (Reimand & Bader, 2013) or interaction interfaces (Porta-Pardo *et al*, 2015; Engin *et al*, 2016). Recently, Vogelstein *et al* (2013) have shown that oncogenes often contain not only regions but also specific residues which accumulate a high fraction of the overall mutational load within a gene. In addition, Chang *et al* (2016) developed a statistical model for detecting residues with a high mutation frequency and applied it to the pan-cancer data.

1 Department of Biology, Institute of Molecular Systems Biology, ETH Zurich, Zurich, Switzerland
2 Division Signaling and Functional Genomics, German Cancer Research Center (DKFZ), Heidelberg, Germany
3 Faculty of Science, University of Zurich, Zurich, Switzerland
4 Department of Cell and Molecular Biology, Faculty of Medicine Mannheim, Heidelberg University, Heidelberg, Germany
5 German Cancer Consortium (DKTK), Heidelberg, Germany
 *Corresponding author. Tel: +41 4463 33170; E-mail: aebersold@imsb.biol.ethz.ch
 **Corresponding author. Tel: +49 6221 421950; E-mail: m.boutros@dkfz.de

Importantly, both approaches demonstrated that the sheer presence of such residues was often sufficient to identify cancer driver genes.

Many of the individual cancer mutations are not well studied in terms of how they influence the properties of proteins (Cancer Genome Atlas Research *et al*, 2013). In addition, due to the artifact-prone raw data and inconsistency in mutation calling, the genome-sequencing information is still noisy (Alioto *et al*, 2015). Accordingly, integration of relevant protein structural and functional annotations with mutational patterns could help in distinguishing variants with a likely impact. Here, based on previous observations (Vogelstein *et al*, 2013), we developed an approach for the detection of single protein residues that accumulate point mutations at a significantly higher rate than their surrounding sequence, which we refer to as "hotspot" residues. Specifically, we used the developed approach to obtain a comprehensive set of such protein residues and investigate protein properties that associate with them. The methodology we used is robust to gene length, background mutation rates, and presence of common variants. We make this tool available as an open-source *DominoEffect* R/Bioconductor software package (Code EV1).

In this study, we applied the tool to 40 cancer types with the TCGA or ICGC sequencing data and identified 180 hotspot mutation residues in 160 genes that had a likely functional impact. These mutations alone had the power to cluster tumors based on the cell type of origin, and many of the hotspots were found within proteins, for example, enzymes, that are known to exist in the active and inactive states. Importantly, we found that a significant fraction of the hotspots resided within tumor suppressors. Furthermore, two-thirds of the identified instances were not classified as known cancer genes but many could be functionally linked to cancer pathways or were, as exemplified by Poly(rC) binding protein 1 (PCBP1), previously suggested to have a role in the regulation of cancer genes and proteins. We next characterized the affected protein regions using sequence annotations and associated data on their structural, functional, and interaction features. These analyses showed that the hotspot residues often fell within regions responsible for binding ligands, nucleic acids, and other proteins. To further follow up on this observation, we used available structural data and homology-based 3D models for human complexes. We mapped protein interfaces in these and based on the presence of mutation clusters within the mapped interfaces, we were able to identify 87 proteins in which cancer mutations were likely to affect protein interactions. Again, two-thirds of the instances were proteins that have not previously been defined as cancer drivers. These, among others, included the coactivator-associated arginine methyltransferase 1 (CARM1) and the retinoid X receptor alpha (RXRA), which both also had hotspot residues. Overall, characterization of the recurrent functional mutations suggests that a disruption and dysregulation of protein interactions could be an important molecular mechanism for switching functions of cancer proteins.

## Results

### Hotspot mutations point to known and candidate cancer driver genes

We collected single nucleotide mutation data deposited as a part of the TCGA and ICGC projects and mapped these nucleotide changes

to the encoded protein sequences using the Ensembl gene annotations (Yates *et al*, 2016; see Materials and Methods). Collectively, the data encompassed 40 different cancer types from 22 tissues, with the sequencing information from ~10,000 tumor samples, including ~1,300,000 mutations within coding sequences (see Materials and Methods).

Next, we developed and applied a tool we term *DominoEffect* (Code EV1). The tool identifies and characterizes individual hotspot protein residues that accumulate mutations at a significantly higher rate than their surrounding protein sequence (Figs 1 and EV1). In

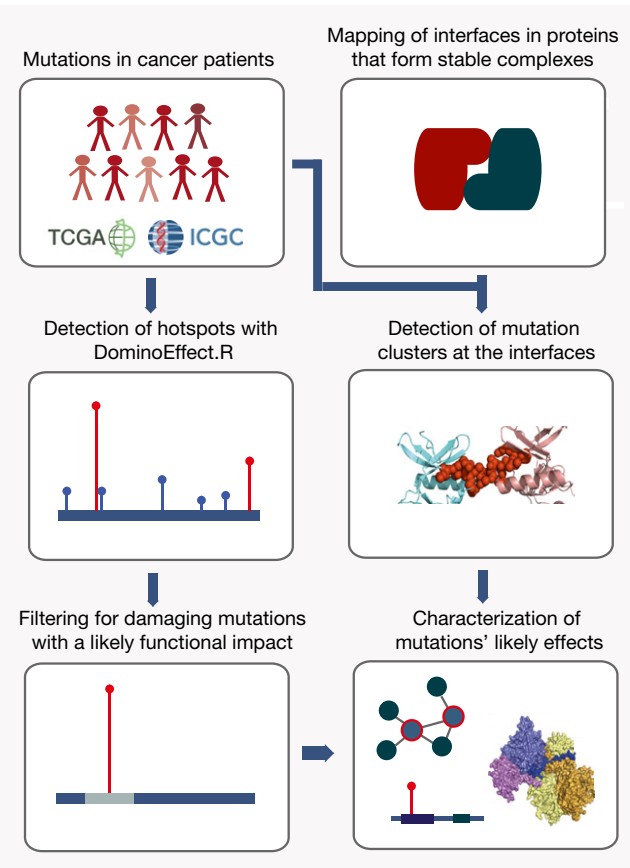

**Figure 1. Outline of the approach.**

Identification of hotspot mutations from the TCGA and ICGC data includes detection of residues that accumulate mutations at a significantly higher rate than their surrounding regions, followed by excluding common polymorphisms and mutations that are not likely to have functional effects. To better characterize protein elements that embed these mutations, we performed an extensive annotation and analysis of both hotspot residues and proteins that contained them. Furthermore, the approach applied to search for interaction interfaces that accumulate cancer mutations at a high rate is shown. This includes collection of structures and structural models for stable protein interactions, mapping of interfaces in these and assessment if cancer mutations are present within the interfaces at a significantly higher rate than elsewhere in the protein sequence. Finally, thus identified proteins and interfaces are further characterized.

this study, we applied *DominoEffect* to search for protein "hotspot" residues that accumulated a high mutation load (here defined as 15% of the mutations) within the windows of 200 and 300 amino acids (see Materials and Methods for the explanation of thresholds). Next, we filtered the obtained residues to avoid false assignments and included only mutations that have a likely functional impact. The largest sources of false positives should be sequencing errors that repeatedly occurred at the same residues or individual polymorphisms, which were not detected in the paired healthy tissue. To account for the latter, but not to exclude the germline risk variants, we filtered out all residues that were common polymorphisms in the human population (i.e., genomic variants with a reported population frequency higher than 1%). We based the filtering on the available data from the 1000 Genomes Project (Genomes Project *et al*, 2015), Kaviar (Glusman *et al*, 2011), Exome Aggregation Consortium (Lek *et al*, 2016), and Ensembl-linked databases (Sherry *et al*, 2001; Yates *et al*, 2016). To further filter out both, sequencing errors and population polymorphisms, we applied the PolyPhen-2 algorithm which assesses a likely mutation effect on the protein function (Adzhubei *et al*, 2010). PolyPhen-2 uses a probabilistic classifier with eight sequence-based and three structure-based features. This filtering step can also exclude true disease hotspots that do not have a sufficient structural or evolutionary support for strong effects. However, to gain more confidence in the individual predictions, we deemed it necessary to account for the substantial presence of false positives in the initial set of hotspot residues. For instance, nearly a third of the initially identified "hotspots" (132 out of 451) were annotated as common variants, which still likely represents an underestimate as a catalog of non-disease human polymorphisms is still incomplete. Simulations of randomly re-assigned mutations within titin, that is, a gene with the highest overall mutation burden, did not report any hotspot residues (1,000 repetitions). Thus, under the naïve assumption that each amino acid is equally likely to be mutated at random, background mutations should not strongly contribute to false positives.

Using the approach introduced here, we applied the *DominoEffect* tool to the pan-cancer data and identified both known instances of hotspot driver mutations as well as residues that were as yet not annotated as such. In total, we identified 180 hotspots within 160 genes (Dataset EV1) for which the reported mutations were categorized as deleterious by the PolyPhen-2. This set, thus, represents frequently mutated residues that could be of particular functional relevance in cancer development. The gene set was also enriched for known cancer drivers (54 or 34% of the genes with hotspots were in the Cancer Gene Census). For a comparison, a fraction of known drivers among the genes that were selected by simply asking for a high mutation load within a protein (more than 100 mutations) or at an individual residue (more than 5 mutations) was 7.9% and 9.6%, respectively. Of note, on average, 88% of tumor allele changes assigned as hotspot mutations were reported as heterozygotic in the TCGA dataset. The most commonly mutated amino acid was arginine (37% of the hotspot residues had arginine as a reference amino acid, while its overall frequency in the reference proteome was 6%). The second most frequently mutated amino acid was glutamate, which was predominantly mutated to lysine (11% of all events). For an illustration, in a protein set with an equal representation of all codons and an equal probability of nucleotide changes, a frequency of such mutation would be only 0.61%. Independently of the hotspot analysis, we additionally searched for cancer mutation clusters in known and modeled protein interaction interfaces (Fig 1).

Strikingly, 36% (3,679/10,118) of the analyzed cancer genomes had at least one of the 180 hotspot residues mutated (Fig 2A). For a comparison, the same size randomly selected gene set that contained any of the protein positions with five or more pan-cancer mutations was on average mutated in 14% of the patients ($P < 6 \times 10^{-12}$, distance from the observed distribution of 1,000 random values). The major contributors to the highly prevalent mutations were the well-studied oncogenes KRAS, BRAF, IDH1, PIK3CA, NRAS, SF3B1, CTNNB1, and PTEN: More than one-quarter (i.e., 27%) of all patients had a hotspot mutation in at least one of these genes. However, in the whole set, a majority (106/160) of the genes with hotspots were not previously annotated as cancer drivers. Importantly, 18 (i.e., 17%) of these candidate genes had a homolog in the Cancer Gene Census (Futreal *et al*, 2004). These instances are listed in the Dataset EV2 and they included, among others, RBL2, KLF5, and ARAF as homologs of cancer drivers RB1, PBX1, and BRAF, respectively. The fraction of Cancer Gene Census homologs in the background set of human genes was significantly lower than among the hotspot genes (7 versus 17%, respectively, $P < 3 \times 10^{-4}$, chi-square test). Therefore, the approach used here is capable of suggesting biologically relevant cases for further follow-up studies, while maintaining an overall low false-positive rate (Marx, 2014).

---

**Figure 2.  Properties of proteins and residues with frequent hotspot mutations.**

A   A high fraction of the sequenced samples (i.e., 36%) have at least one of the detected hotspot residues mutated. Strong contributors to this signal are the listed known cancer drivers with hotspots. Hotspots in one or more of these proteins are mutated in 27% of the analyzed tumor samples.

B   Proteins with hotspot mutations (dark cyan) have a higher fraction of enzymes than other proteins in the Cancer Gene Census (light red, *P* < 0.015) or all other human proteins (dark green, ****P* < $10^{-4}$, chi-squared test). Color representation of proteins from the three sets follows the same scheme on all the following figures.

C   Protein domains that are significantly overrepresented among the proteins with hotspot mutations compared to the non-cancer background proteins (adjusted *P* ≤ 0.01, Fisher's test).

D   Protein annotations (GO terms) that are overrepresented among the proteins with hotspots compared to the non-cancer background proteins (adjusted *P* < $10^{-4}$, Fisher's test). Transcription factor binding and several other terms related to regulation of gene expression are more abundant among the other proteins in the Cancer Gene Census than among the genes with hotspots.

E   Hotspot mutations that mapped to the functionally similar protein segments in two or more different proteins are classified depending on whether the mutation occurred within a protein domain with an enzymatic function (first column), another region in the protein that can mediate binding to other proteins, nucleic acids, ligands or lipids, or in a region that contains a signaling motif or a transmembrane segment.

F   Examples of individual proteins of interest. PCBP1, RXRA, and CARM1 are functionally related to cancer-relevant processes and have hotspot mutations (dark red circles) within the RNA binding KH domain, ligand-binding segment, and enzymatic domain, respectively. Other missense mutations in these proteins are depicted as blue circles. Pfam protein domains encoded by these genes are shown as colored boxes.

## Genes with hotspot residues can often exist in both an active and inactive state

The obtained extensive set of known and candidate cancer-associated genes with hotspot mutations provides an opportunity to define gene and protein characteristics associated with such residues. Mutational impacts are well characterized for the few most frequently mutated cancer hotspots, but general principles of protein properties that associate with hotspot residues have not yet been systematically investigated. From the biological view, hotspot

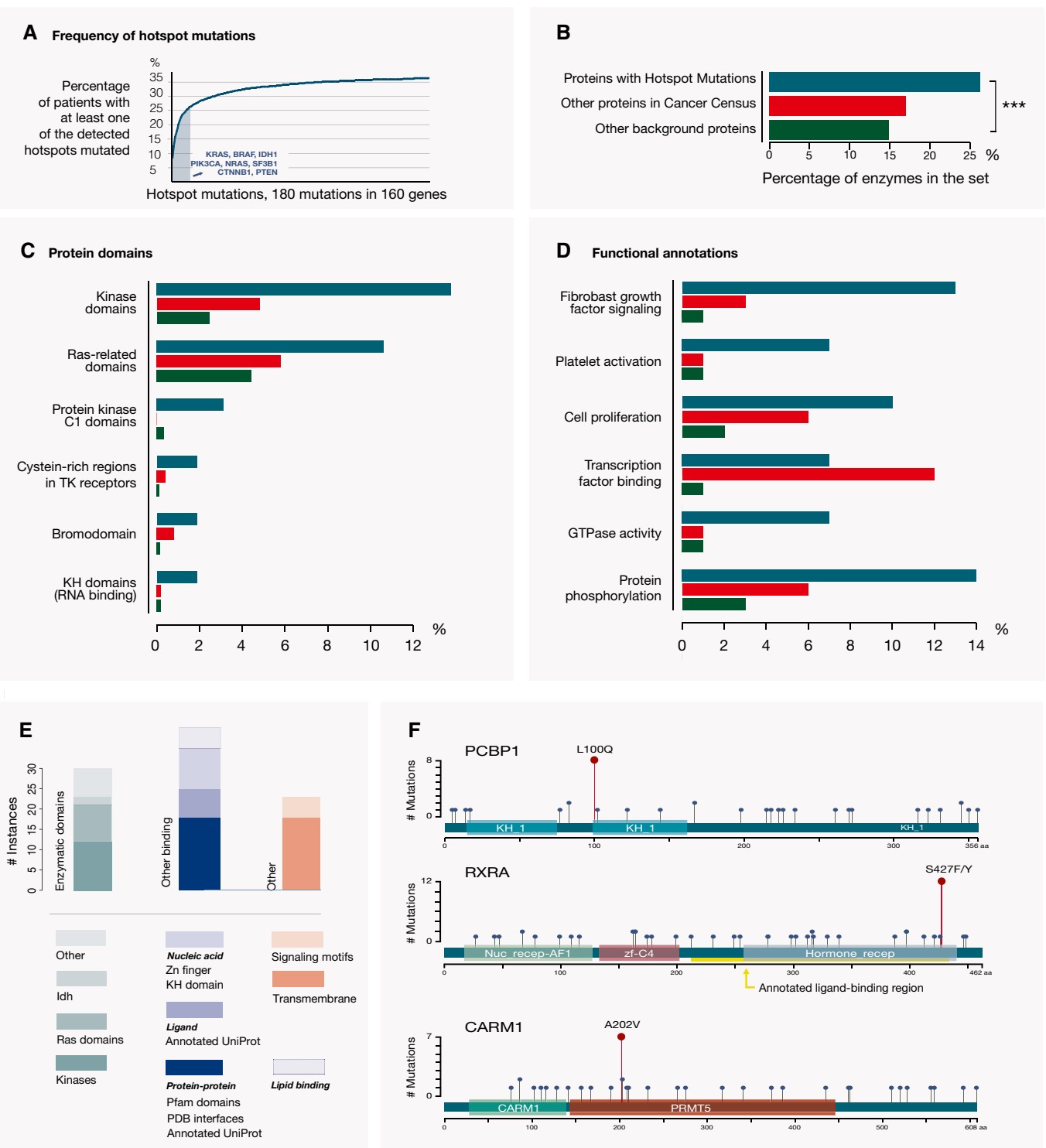

**Figure 2.**

residues should be enriched in single positions with the power, if mutated, to affect the overall protein function. Among the most prominent examples of cancer drivers with hotspot residues are kinases and Ras proteins where the hotspot mutations frequently act by switching the proteins to a constantly active state. Both of these protein families have enzymatic activity, even though Ras proteins need additional activators for this. We hence assessed if the genes with hotspots overall contained a high fraction of enzymes. For this, we used the Expasy Enzyme database annotations (Bairoch, 2000) and compared genes with hotspot residues to the (i) genes in the Cancer Gene Census (excluding the genes with hotspot mutations and homologs of genes with hotspot mutations) and to (ii) all other protein coding genes. We indeed observed that, compared to the other two categories, genes with hotspots were enriched in enzymatic functions (26% of the genes with hotspots were annotated as enzymes; $P \leq 0.015$ and $P < 9.3 \times 10^{-5}$, chi-squared test, respectively, compared to the other two sets; Fig 2B and Dataset EV3). Of note, Expasy Enzyme annotations did not include the Ras proteins, as GTP hydrolysis does not represent their main function (Dataset EV3). The fact that the observed trend was also present in a comparison with the genes in the Cancer Gene Census suggested that the observation should not be influenced by a bias in protein annotations, which are likely to be more complete for better-studied genes. Moreover, the fraction of enzymes was even two times higher among the here-identified genes with hotspot residues than among the genes in the Cancer Gene Census that were involved in translocations ($P < 0.0003$). This suggests that different gene classes could preferentially be activated through different mutational processes (Latysheva *et al*, 2016). To confirm that this observation was not influenced by the filtering of hotspots to include only those with a likely deleterious effect (using PolyPhen-2), we assessed a fraction of enzymes in the initial set of hotspots after excluding only common polymorphisms. In this set, 21% of proteins (i.e., 62 out of 290) were annotated as enzymes. This was a lower fraction than in the final set, but still considerably higher than the fraction of enzymes in the background set of human proteins ($P < 7.6 \times 10^{-5}$, chi-squared test). Of note, a set of the Cancer Gene Census genes without the hotspot genes may still contain drivers with hotspot mutations that have not been detected by the approach applied here.

In accordance with these observations, protein domains that were most frequently encoded by the genes with hotspot residues were kinase and Ras domains, Fig 2C and Dataset EV4; adjusted *P*-values for the overrepresentation were $< 6 \times 10^{-9}$ and $< 0.008$, respectively, Fisher's test, when compared to the background set of non-cancer proteins). In addition, the genes with hotspot residues were also enriched in proteins involved in the related Rho GTPase signaling: 10 genes were associated with this pathway, which represents four times higher fraction than in the set of non-cancer background genes ($P < 10^{-3}$, Fisher's test). Both Rho and Ras proteins are known to often function as binary switches. Among the proteins with hotspots were also those with bromodomains, which are functional elements found in epigenetic regulators, and KH domains that play a role in RNA binding. Both domain types were overrepresented compared to the background set of non-cancer proteins (adjusted $P < 0.03$, Fisher's test). Furthermore, genes with hotspot residues were strongly enriched in the functional categories (GO terms) associated with signal transduction; receptor signaling linked

to the fibroblast growth factors ($P < 10^{-16}$, Fisher's test, compared to non-cancer proteins) and platelet activation (adjusted $P < 10^{-4}$, Fisher's test), a process that is known to play a role in metastasis (Gay & Felding-Habermann, 2011). These processes were also enriched when compared to other proteins in the Cancer Gene Census ($P < 10^{-4}$ and $P < 10^{-3}$, respectively, Fig 2C and Dataset EV4), but several functions associated with cancer development, such as cell proliferation, transcription factor binding, and regulation of gene expression, were enriched only compared to non-cancer background genes ($P < 10^{-4}$). In fact, some of the categories, such as transcription factor and chromatin binding, were more abundant among those Cancer Gene Census genes that were not among the here-identified "hotspot genes" (Fig 2D and Dataset EV4). Overall, this shows that many of the genes with hotspot residues relate to cancer-associated processes and that jointly more than a third of them encode proteins that are known to exist in an active and inactive state.

### Hotspot residues frequently map to crucial functional elements in proteins

Next, we sought to identify which functional elements within protein sequences were common target sites for hotspot mutations. For this, we first mapped annotated functional regions using the Pfam predictions (Finn *et al*, 2016) and UniProt Knowledgebase (KB) annotations (The UniProt, 2017). We then investigated whether the hotspot residues were mapping to the same functional domains in different proteins. We found that domains which were targets of hotspot mutations in two or more proteins were often associated with enzymatic activities, or with binding to proteins, nucleic acids, or lipids (Fig 2E and Dataset EV1). Most frequently, hotspot residues localized within kinase and Ras domains (12 and 9 times, respectively), which also represented a predisposition for domain regions within these protein sequences. Namely, 86% and 100% of the mutations were within the respective domains, which is higher than expected, considering the domain coverage of proteins ($P < 4 \times 10^{-4}$ and $P < 0.09$, chi-squared test, respectively). In addition, hotspot residues mapped to other domains known to be associated with cancer proteins, such as the lipid binding PH domain (Futreal *et al*, 2004), but also to enzymatic and binding domains within proteins that are not yet in the Cancer Gene Census (Dataset EV1). This, for instance, highlighted the Poly(rC) binding protein 1 (PCBP1, Fig 2F) as a possible cancer gene candidate. PCBP1 is as yet not classified as a driver (Futreal *et al*, 2004), but it is involved in the regulation of the expression of a number of cancer genes and has been previously implicated in tumorigenesis and metastasis (Wang *et al*, 2010; Huo *et al*, 2012; Wagener *et al*, 2015; Zhou & Tong, 2015). A hotspot residue within this protein mapped to the KH domain that has a role in nucleic acid recognition and is also a target of a hotspot mutation in the known cancer driver FUBP1.

Further, mapping of hotspot residues onto UniProtKB functional annotations showed that in several instances mutations were falling within protein segments involved in ligand binding. These included both well-known cancer drivers and proteins that are as yet not in the Cancer Gene Census, such as nuclear receptor RXRA (Fig 2F and Dataset EV1). As a following step, we analyzed the PDB structures of proteins with the identified hotspot mutations (Berman *et al*,

2000; www.rcsb.org). When structural data were available, we aligned the segments with hotspot mutations onto the corresponding PDB sequences (see Materials and Methods). Jointly, the analyses of both UniProtKB annotated features and available structures highlighted protein interaction interfaces as a functional element class that was an important target of hotspot mutations (Dataset EV1). These included the well-studied cases of CTNNB1, PIK3R1, and SMAD4 proteins as well as proteins that are as yet not annotated as drivers, but are functionally connected to cancer pathways. An example of this are hotspot residues within the CARM1 and METTL4 proteins that mapped to the interaction interfaces within their protein arginine methyl transferase (PRMT) domains. The PRMT domain is commonly found in epigenetic regulators and mediates methylation of arginine residues on histone tails. The observed hotspot mutations in CARM1 and METTL4 could thus have an effect on the interactions between the CARM1 proteins that together form a complex, or on the METTL4 binding to its interactor protein METTL3. Both METTL4 and CARM1 (Fig 2F) are not yet in the Cancer Gene Census. Finally, hotspot residues in the PBX2 and MAX proteins mapped to their conserved DNA interfaces, suggesting abolished or altered transcriptional regulation. Jointly, the ability of mutations to affect protein interactions or nucleic acid and ligand-binding properties suggests the means by which they could have a potential to switch the protein function and influence downstream processes in the cell.

## An important role of hotspot mutations is the inactivation of tumor suppressors

Tumor suppressor genes can be inactivated during the course of disease either through mutations that affect their transcription levels or through changes that influence protein expression, stability, or function. In the case of the PTEN (Papa *et al*, 2014), FBXW7 (Welcker & Clurman, 2008) and SMAD4 (Miyaki & Kuroki, 2003) genes, as well as several other tumor suppressors (de Vries *et al*, 2002; Hanel *et al*, 2013), a frequent mechanism of inactivation is through point mutations that reoccur at the defined residues and act dominantly on the molecular level. For these genes, disease-causing effects of single inactivating mutations were demonstrated by the follow-up functional studies and in animal models (Taketo & Takaku, 2000; Welcker & Clurman, 2008). In the cancer genomics studies, mutational clustering is often used as a signature that is associated with oncogenes (Davoli *et al*, 2013; Vogelstein *et al*, 2013). However, these individual examples show that hotspots can also pinpoint to a preferential mechanism for tumor suppressor inactivation. In order to categorize hotspot mutations that could function by inactivating tumor suppressors, we analyzed which of the here-identified genes were categorized as tumor suppressors in the Cancer Gene Census, or were predicted to be suppressors based on mutation signatures in an independent study (Davoli *et al*, 2013). In addition, to further expand the set of tumor suppressor candidates, we assessed the frequency of deleterious mutations within the here-defined set of hotspot genes using the mutation data from the above-described TCGA and ICGC datasets. For this, we considered changes with a likely highly deleterious impact on the protein sequence (i.e., premature nonsense mutations and out of frame insertions/deletions), and we compared these to the synonymous mutations within the same

genes. The latter represented neutral mutations and served as a measure of a background mutation rate in the sequences (see Materials and Methods). Using a conservative threshold that required an overrepresentation of deleterious over neutral changes within a protein, we found that 15 genes with a hotspot mutation also exhibited mutation patterns typical of tumor suppressors (Dataset EV5). Collectively, together with the proteins that were annotated as suppressors in the Cancer Gene Census or were suggested as tumor suppressors in a previous pan-cancer study (Davoli *et al*, 2013), this analysis highlighted 23 (i.e., 14%) of the here-identified genes with hotspots that could, at least in some contexts, act as tumor suppressors (Dataset EV5). In addition to the known cancer drivers, this list included ARHGAP5 and RBL2 genes, which are homologs of the SRGAP3 and RB1 Cancer Gene Census genes, respectively (Fig 3A and Dataset EV2). Of note, the obtained list of tumor suppressors with hotspot residues is likely still incomplete. For instance, literature evidence and mutation patterns also highlight the KLF5, SPOP, and POLE genes as possible tumor suppressors even though they do not pass the set thresholds (Fig 3A and Dataset EV5). Overall, this analysis supports the notion that an important fraction of hotspot mutations could act by turning the original protein function off.

Furthermore, to gain additional insights into the distribution of functional effects of hotspot mutations, we classified genes in which hotspot residues were mutated in 10 or more tumor samples into the following three categories: (i) instances where mutations occurred within a tumor suppressor and were likely inactivating, (ii) instances where mutations affected protein binding properties and were likely activating, and (iii) all other instances (Fig 3B and Dataset EV6). These more frequently mutated instances encompassed in total 53 genes and included the experimentally characterized hotspots within the PTEN, FBXW7, SMAD4, EP300, and CREBBP tumor suppressors. In the set of proteins with more frequently mutated and better-characterized hotspots, 32% (i.e.,17 of 53) were tumor suppressor, 51% (i.e., 27 of 53) represented instances where hotspot mutations were likely to affect protein binding properties, and 17% (i.e., 9 of 53) were proteins in which the impacts of hotspot mutations could not be readily classified. Since a considerable fraction of better-characterized hotspots is likely to function by inactivating a tumor suppressor, this should also be considered as one of possible mechanisms of action for the less well-characterized hotspots.

## Proteins with mutation hotspots are often fundamental in cancer interaction modules

It was suggested that proteins involved in cancer should play a central role in cellular interaction networks (Davoli *et al*, 2013). However, proteins interesting from a disease perspective are often more intensively studied and, as a result of this, a larger number of their interaction partners are catalogued compared to proteins with an unknown function. Using a complementary approach, we compared the presence of iPfam domains (Wang *et al*, 2012; Finn *et al*, 2014), that is, Pfam domains that can mediate protein–protein interactions, in the protein sets that we established above: (i) proteins that contain hotspot mutations, (ii) other proteins encoded by the Cancer Gene Census (described above), and (iii) all other human proteins. The catalog of iPfam domains is built on the

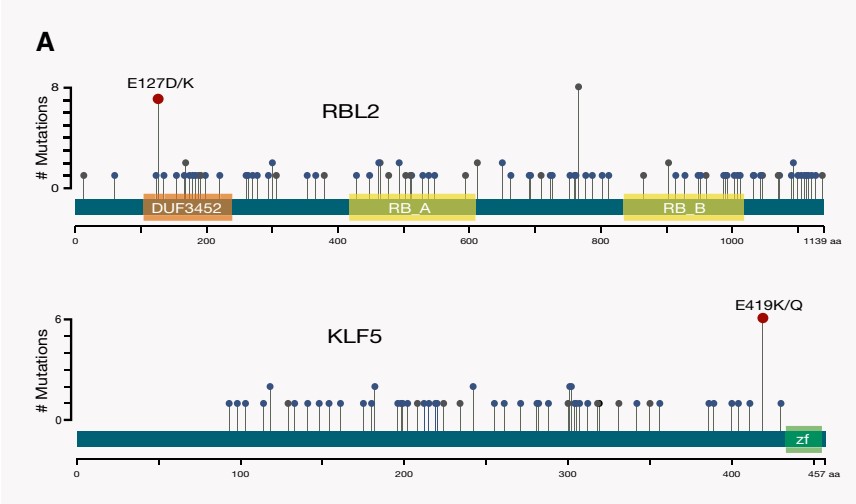

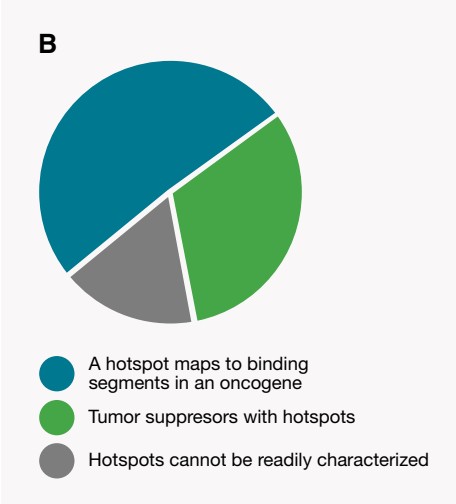

**Figure 3.** **An important fraction of hotspot mutations has a role in inactivating tumor suppressors.**

A   A number of hotspot mutations are associated with tumor suppressors. Among these are RBL2 and KLF5 proteins that are not in the Cancer Gene Census, but are homologs of cancer drivers. Hotspot mutations within these proteins are shown as dark red circles, while missense and highly deleterious mutations are depicted as blue and gray circles, respectively. The encoded Pfam domains are shown as colored boxes.

B   Almost a third (i.e., 32%) of the genes with better-annotated, more common hotspot mutations (observed in at least ten patients) are tumor suppressors. Another prominent category is genes with apparently activating mutations that affect binding properties of oncogenes (51% of the total 53 genes).

available PDB structures, but using sequence features should generally be less strongly affected by biases in previous knowledge than the reliance on targeted interaction data. We observed that Pfam interaction domains were more abundant in the two sets of cancer-associated proteins, than in the background human proteins ($P < 10^{-4}$ and $P < 10^{-13}$, respectively, chi-squared test), whereas no significant difference between the first two sets was observed, Fig 4A. Hence, this underlined that forming interactions with other proteins was often an essential part for the function of proteins with hotspots.

Proteins with similar functions are often close in interaction networks, and functional and disease annotations of a protein's interaction partners can assist in determining its role in the cell (Lundby *et al*, 2014; Menche *et al*, 2015). In accord with this, closeness to modules with known cancer drivers could assist in identifying less well-studied proteins that could be functionally related to cancer pathways. To address this, we searched interactions deposited in the BioGRID database (Chatr-Aryamontri *et al*, 2015) classified as physical associations or direct interactions, and focused specifically on the 106 proteins with hotspots that were not in the Cancer Gene Census. We found that first interaction neighbors of these proteins were indeed enriched in Cancer Gene Census proteins (13% versus expected 3%, $P < 10^{-166}$, when compared to random networks, Fig 4B and C). Individually, 20 (i.e., 19%) of these 106 proteins had interaction neighborhoods strongly enriched in the known cancer drivers (adjusted $P < 0.05$, Dataset EV7), and these clusters often included other proteins with hotspot mutations (Dataset EV7). Of note, known cancer drivers have on average a higher number of interaction partners than other proteins so their presence in the interaction neighborhood is not a direct evidence for the protein's association to cancer. However, it can be useful for better understating of protein roles in the cell. The obtained

interaction network for the 19 proteins is shown in Fig 4D. This included KLF5 and SOS1 proteins, which are both homologs of known cancer driver genes (Dataset EV2), as well as transcriptional regulators CARM1 and ZBTB7A, and signaling regulators RHPN2 and GPX1. Additionally, 27 interaction pairs in which one partner was a candidate with a hotspot mutation and the other partner was a Cancer Gene Census protein were also supported with a PDB structure or a homology-based structural model that indicated a stable interaction between the proteins (see Materials and Methods). Notably, the RXRA protein was in protein complexes with RARA, PPARG, NCOA1, NCOA2, and NCOR2 cancer drivers. All of these complexes have been described at the structural level. The hotspot mutation in this gene was originally reported in the TCGA study of bladder cancer (Cancer Genome Atlas Research, 2014a), but its impact has as yet not been characterized. The observation that a number of the candidates with hotspot mutations were physically associated with each other or with known cancer drivers suggests their possible connections to interaction networks relevant in disease development.

### Recurrent mutations group cancer types by their cell type of origin

We further assessed whether individual hotspot residues were mutated across different tumor types, or whether they were more often specific for a single origin of tumor. To address this, we first grouped similar tumor types from the TCGA and ICGC datasets (see Materials and Methods) and analyzed how a pan-cancer signal for each hotspot was distributed across the different tumor types. We found that 35 out of 180 hotspot residues were largely associated with one tumor type, that is, more than 75% of the mutations on these 35 positions were reported in a single tumor type (Dataset

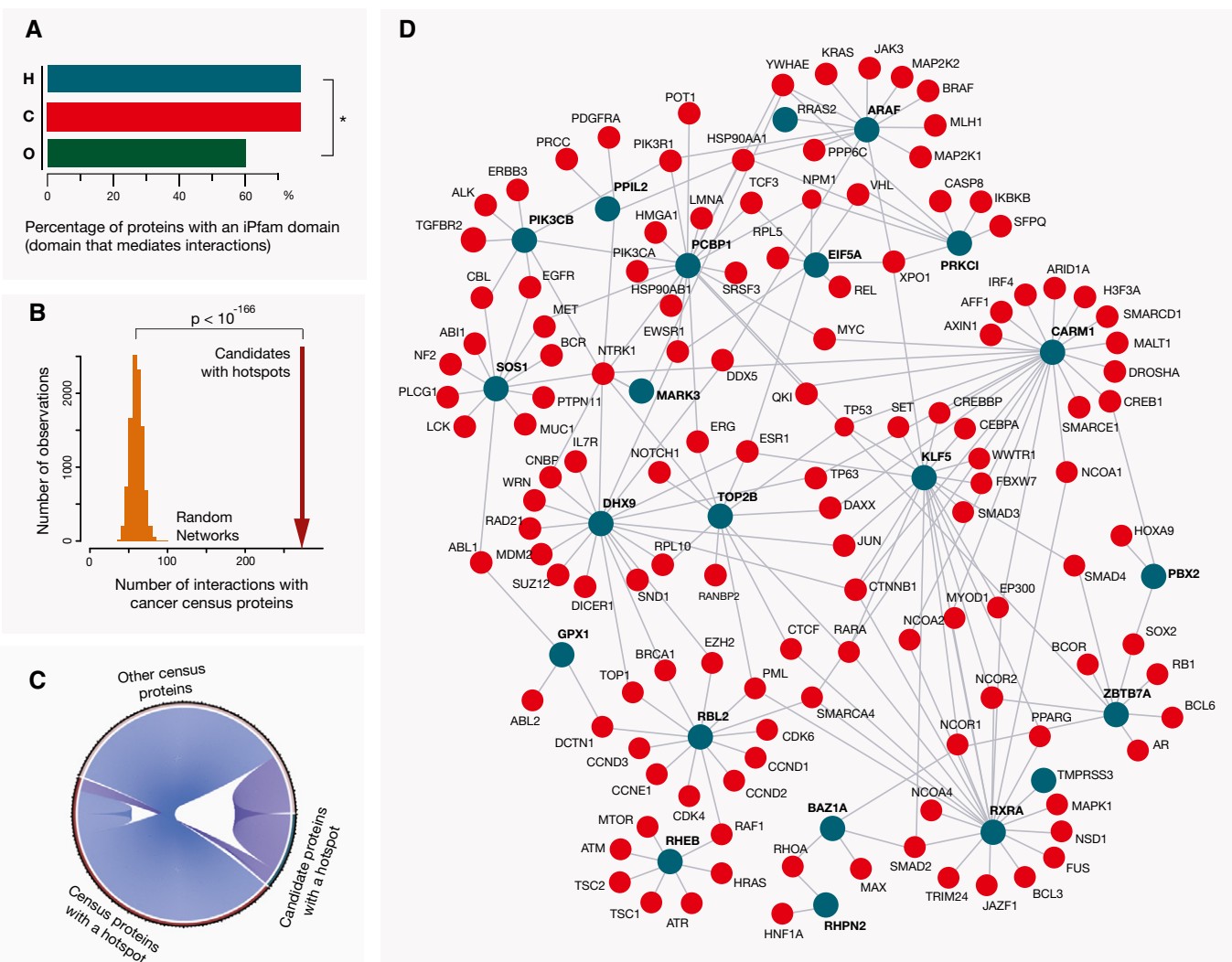

**Figure 4. Proteins with hotspot mutations are often in the interaction networks with known cancer driver proteins.**

A   A fraction of genes with hotspot mutations (H) that encode interaction domains is comparable to the fraction of genes in the Cancer Gene Census that encode the same domains (C) and is significantly higher (*$P < 10^{-4}$, chi-squared test) than the fraction of other background genes (O), with these domains.

B   Direct interaction partners of candidate cancer proteins with hotspot mutations (i.e., those that are not in the Cancer Gene Census) are significantly enriched in known cancer drivers. Distribution of values for the random networks of the same size is shown as an orange histogram (1,000 networks obtained with resampling of the interactors), and the observed physical interactions between the hotspot candidates and Cancer Gene Census proteins (i.e., 273) are indicated with a dark red arrow ($P < 10E-166$, pnorm test).

C   Candidate cancer proteins with a hotspot mutation often interact both with cancer drivers with a hotspot mutation as well as with other Cancer Gene Census proteins.

D   Candidate proteins with hotspot mutations whose interaction neighborhoods were enriched in cancer driver proteins (adjusted $P < 0.05$, Fisher's test) are shown as dark cyan circles and denoted in a bold font. Their Cancer Gene Census interaction partners are shown as red circles.

EV8). This included SPOP, which was exclusively mutated in prostate adenocarcinoma, and RXRA, where 80% of mutations were observed in the bladder urothelial cancer. However, a larger number of positions were mutated in multiple tumor types. For instance, for 89 hotspot positions, a single tumor type never contributed to 50% or more of the mutations (Dataset EV8).

Next, we performed a hierarchical clustering of the analyzed tumor types, based on the percentage of patients that had a mutation in each of the identified hotspots. Patterns of hotspot mutations could also point to the shared and cell-type-specific pathways

in cancer. To account for the uncertainty in clustering, we used the pvclust R package, as this method includes bootstrap resampling of the detected clusters (Suzuki & Shimodaira, 2006). For the visualization, we produced a heatmap with thus obtained tumor clusters and included only hotspots that were mutated in 33% or more of the patients in at least one tumor type (Fig 5). This showed that many of the detected clusters represented well the cancer etiology. For instance, the majority of adenocarcinomas (ovarian, colorectal, lung, and pancreatic) were clustered together, while squamous cell carcinomas (lung, cervical, and head and

neck) were in a separate cluster. The example of the lung cancer reinforced the notion that a cell type of cancer origin (adenoma versus sarcoma) was more strongly defining the mutation hotspot pattern than a tissue. Uterine carcinomas which can have both epithelial and sarcoma components were in a larger cluster that contained both epithelial adenocarcinomas and sarcomas.

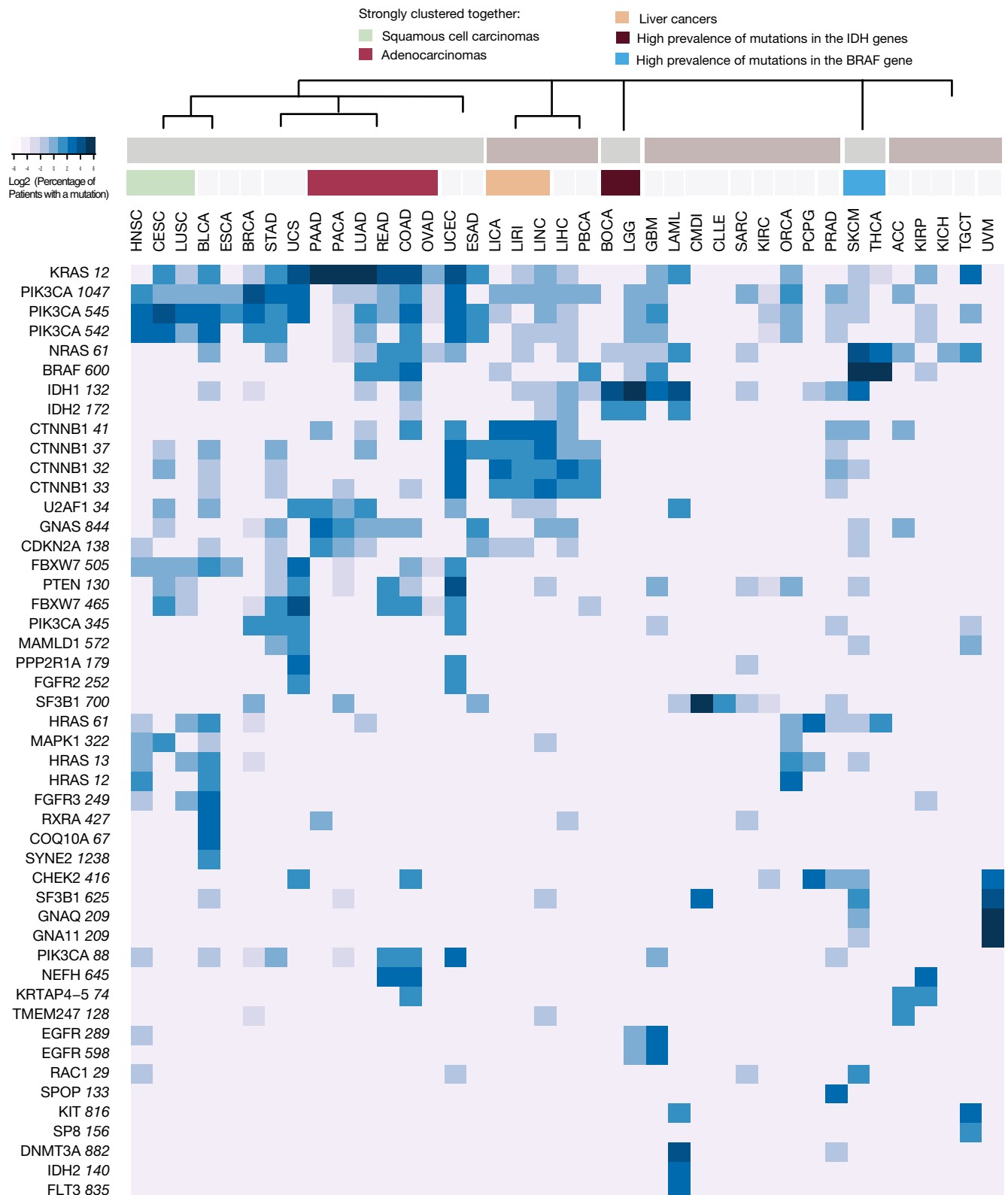

**Figure 5.**

Additionally, liver cancers were clustered together, but kidney tumors of different origins did not show such a clear trend. Blood tumors were largely defined by the scarcity of point mutations, and skin cutaneous melanoma and thyroid cancer were clustered together based on a high frequency of the BRAF hotspot mutation. We included here only pan-cancer hotspots, but those identified on the level of individual tumor types could also contribute to better definition of relationships among the cell types of origin. Overall, this analysis showed a strong power of hotspot mutations to reflect the identity of a tumor of origin.

**Interface mutation clusters reveal disease-relevant interactions**

Characterization of functional protein segments affected by the hotspot mutations has recognized binding interfaces as frequent mutation targets (Figs 2 and 4). However, by focusing solely on single residues that stand out from their surrounding regions, one might miss clusters of adjacent mutations that only cross the significance threshold in combination. To address this, we implemented a workflow in which we investigated the presence of such clusters within the protein interaction interfaces. For this, we used available structures or homology-based structural models (Mosca *et al*, 2013) for protein pairs that were predicted to be part of the same protein complex or form stable interactions (see Materials and Methods). From the representative structures, we extracted information on protein interfaces and assessed whether cancer mutations occurred at a much higher rate within the interaction interfaces than in the rest of the sequence (see Materials and Methods). For the latter, we adapted the ActiveDriver tool (Reimand & Bader, 2013), which was initially developed to assess mutation rates in phosphosites by implementing a logistic regression model.

Proteins in which the mutation clusters were identified with the most significant *P*-values were cancer drivers for which it was previously shown that mutations in their interfaces can lead to disease development by disrupting interactions with regulatory proteins (PIK3CA and PPP2R1A) or by preventing GTP hydrolysis (KRAS, HRAS, and GNAS); Dataset EV9, 87 proteins in total. Importantly, this analysis additionally identified a number of instances that are not in the Cancer Gene Census yet (Table 1), including RXRA and CARM1 (Fig 6A and B). Both RXRA and CARM1 were among the above-identified "hotspot genes", and their hotspot residues were within the interaction interfaces. We further used protein GO term annotations (Harris *et al*, 2004) to assess whether proteins with

interface mutation clusters were enriched in any functional categories. Cellular processes that were overrepresented among these proteins included proliferation, ubiquitin signaling, and complement pathway activation (Dataset EV10, adjusted *P* < 0.01). Ubiquitin-regulated signaling was represented with several relevant examples, most notably with the oncogene CBL and with the CUL1 and CUL4B homologs that are not yet in the Cancer Gene Census. CUL1 plays a role in different cancer signaling cascades and in cell cycle progression (Nakayama & Nakayama, 2006). Of interest, both CUL1 and CUL4B had a mutation cluster at one of the interface contacts with the regulator protein CAND1 (Fig 6C). In addition, we used annotations of protein complexes from the ConsensusPathDB (Kamburov *et al*, 2013) and found a number of protein complexes relevant in cancer that would be affected through these mutations (Dataset EV10). Notably, heterotrimeric G protein complex had one of the strongest signals. The complex is composed of the oncogene GNAS, and the GNB1 and GNG2 proteins (G protein β and γ subunits, respectively). Of relevance, mutations in the GNB1 protein interface were recently shown to promote cellular growth and transformation, as well as kinase inhibitor resistance (Yoda *et al*, 2015). Some of the identified mutation clusters, including those within the GNB1 and GNG2 proteins, had a relatively low mutation count (Dataset EV9). This reflects the fact that often there are multiple options for activating or inactivating relevant cellular pathways and interaction modules (Cancer Genome Atlas Research N, 2014b), which is also one of the explanations of a long tail of cancer driver mutations.

In addition to finding mutation clusters within protein–protein interfaces, the analysis identified a mutation cluster at the interface of the PAX5 transcription factor with DNA, as well as a number of contacts with small ligands, most commonly ATP and GTP. PAX5 is a known tumor suppressor with a highly conserved DNA-binding motif (Garvie *et al*, 2001) that is often involved in leukemia development, however, most frequently through translocations. Overall, mutation clusters in structural interfaces were able to highlight additional cancer-relevant proteins and protein regions, which can mediate a switch in protein activity and function.

## Discussion

Even with the abundance of cancer genomics data, discerning driver and passenger mutations remains a major task (Glusman *et al*, 2017). Nevertheless, examples of driver genes from the commonly

**◄**

**Figure 5.  Hotspot mutations drive clustering of tumor types.**

The depicted heatmap includes hotspots that were mutated in at least one-third of the patients in one or more tumor types. The coloring corresponds to the log2 value of the percentage of patients with a mutation in each tumor type. The range goes from 0 to 78% of patients. The latter value corresponds to the percentage of patients with the KRAS 12 mutation in pancreatic cancer (which is 6.28 on the log2 scale). Clustering of tumor types was performed independently with the pvclust method and it included data for all hotspot mutations. The ordering of tumor types on the heatmap is based on the dendrogram obtained with the pvclust method. A simplified version of the dendrogram above the heatmap indicates clusters that were reported up to the height of 0.5 (the full scale was from 0 to 0.7). In addition, the most distinct individual clusters (corresponding to the height of 0.25) are colored and annotated. Different hotspots in the same gene tend to cluster together, and the clustering of tumor types is largely based on the cell type of origin. Abbreviations: head and neck squamous cell carcinoma (HNSC), cervical squamous cell carcinoma and endocervical adenocarcinoma (CESC), lung squamous cell carcinoma (LUSC), bladder urothelial carcinoma (BLCA), esophageal cancer (ESCA), breast invasive carcinoma (BRCA), stomach adenocarcinoma (STAD), uterine carcinosarcoma (UCS), pancreatic adenocarcinoma (PAAD), pancreatic cancer (PACA), lung adenocarcinoma (LUAD), rectum adenocarcinoma (READ), colon adenocarcinoma (COAD), ovarian serous cystadenocarcinoma (OVAD), uterine corpus endometrial carcinoma (UCEC), esophageal adenocarcinoma (ESAD), liver cancers (LICA, LIRI, LINC), liver hepatocellular carcinoma (LIHC), pediatric brain cancer (PBCA), bone cancer (BOCA), brain lower grade glioma (LGG), glioblastoma multiforme (GBM), acute myeloid leukemia (LAML), chronic myeloid disorders (CMDI), chronic lymphocytic leukemia (CLLE), sarcoma (SARC), kidney renal clear cell carcinoma (KIRC), oral cancer (ORCA), pheochromocytoma and paraganglioma (PCPG), prostate adenocarcinoma (PRAD), skin cutaneous melanoma (SKCM), thyroid carcinoma (THCA), adrenocortical carcinoma (ACC), kidney renal papillary cell carcinoma (KIRP), kidney chromophobe (KICH), testicular germ cell tumors (TGCT), and uveal melanoma (UVM).

**Table 1.  Significant gene hits with mutation clusters at the interaction interfaces that are currently not in the Cancer Gene Census.**

| Gene name | P-value |
|-----------|---------|
| CARM1 | 3.43E-09 |
| C5 | 6.05E-07 |
| METTL14 | 1.30E-06 |
| CUL1 | 2.83E-06 |
| ABCD1 | 1.67E-05 |
| SCN2A | 1.82E-4 |
| IL1RAP | 2.07E-4 |
| MAP1LC3A | 3.66E-4 |
| CNOT1 | 3.66E-4 |
| ITGAX | 0.001 |
| CUL4B | 0.001 |
| UCHL1 | 0.001 |
| SUPT5H | 0.002 |
| BMP2 | 0.002 |
| CSF2RB | 0.002 |
| GNB1 | 0.002 |
| LRRC4C | 0.002 |
| C6 | 0.003 |
| WDR5 | 0.003 |
| POLR1A | 0.003 |
| ILK | 0.004 |
| RXRA | 0.004 |
| FLT1 | 0.005 |
| UBA3 | 0.005 |
| IL6 | 0.006 |
| PRKAG3 | 0.007 |
| TLR4 | 0.009 |
| MCM6 | 0.009 |
| TNPO1 | 0.010 |
| CYFIP1 | 0.011 |
| ACTB | 0.012 |
| CHRNB2 | 0.013 |
| MPO | 0.013 |
| MCM7 | 0.013 |
| SERPINA10 | 0.014 |
| EIF5A | 0.014 |
| KPNA1 | 0.014 |
| CDC7 | 0.015 |
| ITGA2B | 0.019 |
| F10 | 0.019 |
| IRF7 | 0.019 |
| RPA1 | 0.019 |
| CHP2 | 0.020 |
| RAMP2 | 0.024 |

**Table 1** (continued)

| Gene name | P-value |
|-----------|---------|
| COL4A2 | 0.024 |
| GNG2 | 0.025 |
| CAB39 | 0.026 |
| C3 | 0.028 |
| CFB | 0.030 |
| TAF1 | 0.031 |
| KDM1B | 0.034 |
| TRAF2 | 0.034 |
| SCAF8 | 0.037 |
| MAPK8 | 0.037 |
| IL21 | 0.037 |
| XRCC5 | 0.045 |
| TTC7B | 0.046 |
| PSMB3 | 0.046 |

mutated protein families, such as Ras proteins and kinases, do show that similar molecular principles are often reused for the activation of oncogenic processes. As a corollary, comprehensive cataloging of residues, which can facilitate a switch in protein function, could help in identifying additional cancer-associated genes. Here, based on the sequencing data for more than 10,000 cancer genomes, we composed a set of most common hotspot mutations with a likely functional effect and characterized these using available structural and interaction data, as well as protein sequence features and residue annotations. Frequently, these mutations were located in the protein regions that interacted with ligands, or at the interaction interfaces with other proteins. This represents known mechanisms of cancer development, where the altered ligand binding on one hand, or escape from the regulation by interacting proteins on the other hand, leads to oncogenic activation. The former is illustrated with the BRAF, KRAS, and IDH1 and the latter with the PIK3R1, CTNNB1, and CDK6 proteins. Recognizing which protein segments are particularly important for disease development can be a starting point for more focused studies of these, as we demonstrated here with interaction interfaces. Such "hypothesis-driven" analyses will have more power to detect functional elements mutated at a lower frequency, compared to methods that do not include any biological information.

An additional important role of hotspot mutations is likely to be in inactivating protein function. This class of hotspot residues can hence indicate the most efficient mechanisms for shutting down tumor suppressors. On the cellular level, such mutations can (i) act in a dominant negative manner with respect to the other protein copy (Welcker & Clurman, 2008; Blattner *et al*, 2017), (ii) be sufficient for inactivation if the gene is dosage sensitive, or (iii) be combined with an inactivating mutation in the other gene allele. Mechanistically, these mutations tend to disrupt the original protein function, exemplified by the disrupted substrate binding in CREBBP and EP300, or to prevent the activation of a tumor suppressor, exemplified by mutations within the SMAD4 interface or CHEK2 activation loop (Dataset EV5). Alternatively, these mutations can

**A** RXRA (brown) with PPARG (blue),
NCOA2 peptide, retinoic acid and DNA

**B** CARM1 dimer

**C** CAND1 (purple) and CUL1(blue)
with RBX1 (yellow)

**Figure 6.  Mutations at the interaction interfaces can affect regulation of cancer-associated processes.**

A  Residues within the interaction interface of the RXRA protein (brown) toward the PPARG receptor (peroxisome proliferator-activated receptor gamma, blue) are shown as spheres. A signal for this interface was driven by the hotspot mutation in RXRA (shown as a red sphere). Other protein regions in the representative PDB structure (3dzy) are shown as ribbons. The NCOA2 coactivator peptide is not visible from the shown angle, while the retinoic acid can be partly seen above the interface and it is represented with dark red sticks.

B  CARM1 protein dimer is shown within the representative PDB structure (5dx0). Again, the interface segments are depicted as spheres and the rest of the protein as ribbons. Significant signal within the interface is driven by the CARM1 hotspot mutation that is shown in red.

C  Interaction interfaces that gave a strong signal for the clustering of cancer mutations in the CAND1 (cullin associated and neddylation dissociated 1, purple) and CUL1 (culling homolog 1, blue) proteins are shown as spheres and other protein segments as ribbons. The shown PDB structure (1u6g) represents a complex that additionally contains the RBX1 protein (RING-box protein 1, yellow).

interfere with the oncogene regulation, exemplified by the PIK3R1 interface to PIK3CA. Analogously to this, oncogenes that are dosage sensitive are likely to be activated through changes that will result in the increased transcript and protein levels. Of note, some of the genes with hotspots are also frequently found overexpressed in different tumor types (Dataset EV11). Overall, molecular principles of activation and inactivation of genes in cancer will frequently be the same as in other diseases, and interpretation of genomics data in general would be strongly empowered with a comprehensive catalog of functional, structural, and interaction properties of individual residues in human proteins.

As more cancer genomes get sequenced, we expect that the list of genes with hotspot mutations will expand. Furthermore, greater sample sizes can allow for a detection of residues that are mutated only in certain tumor types. Many of the proteins with hotspot mutations identified here do not have well-characterized functions in the cell. In this context, it is interesting that some of the less-studied genes with hotspot mutations were proposed to be important in cellular decision-making in embryonic stem cells (ESCs). A recent screen for the genes that contribute to the commitment of ESCs to differentiation (Leeb *et al*, 2014) highlighted as hits the PRKCI kinase, the DHX9 RNA binding protein, and the ZNF706 zinc finger protein, which were all identified here through their hotspot residues. Notably, retinoic acid receptors are known to play a key crucial role in cell differentiation, which again highlights the example of the RXRA protein (Fig 6A).

Cancer classification based on molecular data, which include point mutations, copy number variations, and mRNA and protein expression, has proven to be able to recapitulate pathological subtypes and suggest finer subclasses within tumor types (Hoadley *et al*, 2014). Remarkably, here we observed that clustering of tumor types based on a small number of functional hotspot mutations was able to largely mimic the behavior of much more complex molecular datasets. This observation thus suggests a central role of hotspot mutations in shaping downstream mRNA and protein expression levels and cellular phenotypes. Of note, hotspots with a high mutation frequency defined the major trends among the clusters, but adding information on more residues was able to refine the relationships among different tumor types. It is conceivable that additional hotspots detected at the level of individual tumor types will also be informative for this. However, the power to detect these hotspots strongly depends on the cohort sizes. A higher number of sequenced genomes should hence allow for a better granularity in discerning cell-type-specific signaling networks, but also aid in recognizing the shared mechanisms of cancer progression.

The hotspot residues identified here encompass only 180 amino acids and represent a very small fraction of the coding genome. Still, more than a third of the sequenced patients had a mutation in at least one of these residues. Of relevance, many (i.e., 45) of the proteins with hotspot mutations are classified as druggable (Dataset EV12). Among these are the mentioned RXRA and PRKCI proteins. Importantly, hotspot mutations can be of a high clinical relevance; they are attractive therapeutic targets, both for small molecules and for antibodies, as well as for immunotherapy. The former is exemplified with the BRAF, KIT, and EGFR proteins. Furthermore, these mutations can be useful in diagnostics, either in gene panels or in liquid biopsies. Hotspot mutations are also of high interest in biomedical studies, as reducing the number of candidate mutations for the follow-up studies simplifies experimental design and reduces costs.

Here, using several resources, we compose a set of human protein complexes and incorporate structural data to analyze these.

In this way, we detect a number of additional interaction interfaces that are significantly affected by cancer mutations. Of interest, many of the identified candidate proteins belong to protein families that already have cancer-associated members (such as MCM6 and MCM7 proteins, MAPK8, or CDC7, Table 1). A larger set of interaction pairs with structural models will likely identify many more cases where formation or dissociation of specific protein interactions has an important role in cancer. Finally, integration of cancer genomics analyses with large-scale experimental approaches, such as functional genomics screens (Boutros *et al*, 2015), protein interaction (Aebersold & Mann, 2016), and genetic epistasis assays (Laufer *et al*, 2013) as well as studies of mutation effects on pathway activation and proteome re-organization (Collins *et al*, 2013), can additionally help in narrowing down medically and clinically relevant candidates. Eventually, a larger catalog of cancer-associated mutations further characterized *in vitro* and in model organisms would be invaluable for understanding different routes of disease emergence and for identifying therapeutic opportunities.

# Materials and Methods

### DominoEffect R package

In order to automate detection and annotation of hotspot mutations from the sequencing data, we developed an R/Bioconductor package *DominoEffect*. The package relies on the gene and protein annotations from the Ensembl database, which can also be obtained through the BiomaRt R package (Kinsella *et al*, 2011). Its central part is an algorithm for the detection of significant hotspot mutations in the protein segments of a defined length. For this, the default is to use windows of 200 and 300 amino acid lengths that are centered on the potential hotspot residues (i.e., residues with 5 or more mutations) and require that in both cases at least 15% of mutations in the segment fall on the same residue. The length of windows was chosen so to encompass an individual protein domain. Domains are frequently independent functional and structural protein units and are typically 100–250 residues long (Chothia *et al*, 2003). A threshold for the fraction of mutations was chosen after assessing different values for the pre-defined window length (i.e., both 200 and 300 aa) and asking for the resulting gene set to be enriched in the genes involved in cancer pathways. The threshold of 15% was associated with the lowest *P*-value for the enrichment in "Pathways in cancer" according to the KEGG pathways annotations (Kanehisa *et al*, 2016). The minimum number of mutations (default 5), window lengths, and thresholds for the required fraction of mutations can be easily adjusted in the package. In addition, the percentage of mutations can be replaced with a requirement for the high overrepresentation of mutations at a candidate hotspot residue when compared to the overall mutation rate in the window. The tool excludes common population variants, gives a hotspot *P*-value (adjusted Poisson test), and overlaps the hotspot residues with the instantly downloaded functionally annotated regions in the UniProt/Swiss-Prot KB. For the latter step, when transferring SwissProt annotations, it controls for the sequence agreement between the UniProt and Ensembl protein sequences. In this study, we used the package on the Pan-cancer data, but with more permissive thresholds, it can also be used for finding hotspot

mutations relevant for individual tumor types. With the default settings, only highly frequent mutations will be identified in the smaller patient cohorts. Details and examples are provided in the package vignette. In addition to cancer, the tool can be applied to any other disease for which sufficient exon sequencing data are available (Hoischen *et al*, 2014).

### Mapping genomic mutations onto protein sequences

The results shown here are in part based upon data generated by the TCGA Research Network: http://cancergenome.nih.gov/. We obtained whole exome somatic mutation data deposited in the TCGA and ICGC cancer genomics repositories and available in September 2015. We retrieved somatic mutations sequencing level 2 files from the TCGA web service and proceeded with the analyses of those files that passed the Broad Institute quality filters and were listed on their MAF Dashboard site (https://confluence.broadinstitute.org/display/GDAC/MAF + Dashboard). This included the following TCGA tumor types: acute myeloid leukemia (LAML), adrenocortical carcinoma (ACC), bladder urothelial carcinoma (BLCA), brain lower grade glioma (LGG), breast invasive carcinoma (BRCA), cervical squamous cell carcinoma and endocervical adenocarcinoma (CESC), cholangiocarcinoma (CHOL), colon adenocarcinoma (COAD), glioblastoma multiforme (GBM), head and neck squamous cell carcinoma (HNSC), kidney chromophobe (KICH), kidney renal clear cell carcinoma (KIRC), kidney renal papillary cell carcinoma (KIRP), liver hepatocellular carcinoma (LIHC), lung adenocarcinoma (LUAD), lung squamous cell carcinoma (LUSC), lymphoid neoplasm diffuse large B-cell lymphoma (DLBC), ovarian serous cystadenocarcinoma (OV), pancreatic adenocarcinoma (PAAD), pheochromocytoma and paraganglioma (PCPG), prostate adenocarcinoma (PRAD), rectum adenocarcinoma (READ), sarcoma (SARC), skin cutaneous melanoma (SKCM), stomach adenocarcinoma (STAD), testicular germ cell tumors (TGCT), thyroid carcinoma (THCA), uterine carcinosarcoma (UCS), uterine corpus endometrial carcinoma (UCEC), and uveal melanoma (UVM).

Additionally, we downloaded files with simple somatic mutations from the ICGC data release 19. We excluded files with an unusually high mutation rate and files that were also available as a part of the TCGA set. This added further mutation data for the following tumor types: acute lymphoblastic leukemia (ALL), bladder cancer (BLCA), bone cancer (BOCA), breast triple negative/lobular cancer (BRCA), chronic lymphocytic leukemia (CLLE), chronic myeloid disorders (CMDI), colorectal cancer (COCA), esophageal adenocarcinoma (ESAD), esophageal cancer (ESCA), ewing sarcoma (BOCA), gastric cancer (GACA), liver cancers (LICA, LIRI, LINC, LIAD, LIHM), lung cancer (LUSC), malignant lymphoma (MALY), neuroblastoma (NBL), oral cancer (ORCA), ovarian cancer (OV), pancreatic cancer (PACA), pancreatic cancer endocrine neoplasms (PAEN), pediatric brain cancer (PBCA), prostate adenocarcinoma (PRAD), early onset prostate cancer (EOPC), and renal cancers (RECA).

To consistently map these mutations onto protein sequences, we downloaded gene and transcript annotations from the Ensembl database (Yates *et al*, 2016), release 73. For each gene, we chose a representative, that is, longest protein coding transcript and used an in-house Perl script to map the genomic mutations to the corresponding amino acids. For the genomic mutation coordinates that were not aligned to the Human reference genome assembly build 37 (i.e., GRCh37), we used the UCSC LiftOver service (Kent *et al*, 2002)

and transferred these to GRCh37. Jointly, the analyzed TCGA and ICGC mutation datasets covered 40 different tumor types which spanned 23 tissues of origin. Sequencing data reported mutations for 10,118 samples in total. In the obtained cancer genomes, more than a million single nucleotide somatic changes within protein sequences were observed (1,356,533 single nucleotide mutations). Seventy percent of these mutations (947,359) categorized as non-synonymous and were predicted to result in an amino acid change.

### Detection of hotspot mutations with a likely functional effect

We developed the R package *DominoEffect* that is described above and that analyses mutation patterns within protein sequences and finds hotspot residues which accumulate mutations at a much higher rate than their neighborhood protein regions. We ran the package on the compendium of the TCGA and ICGC data described above and additionally on the TCGA data alone. The latter was done to account for a possible difference in quality between the files: The downloaded TCGA files passed a quality check for entering the Broad firehose, but the individual ICGC files did not have a reference for the quality. Lower quality of mutation data in some of the files could thus result in the general background mutation noise and prevent detection of the less frequent hotspots. However, merging the two datasets gives more power to detect less prominent thresholds. Hence, we combined the significant results from the both analysis.

A major source of false positives in the set of the identified hotspot mutations is due to common variants that, incorrectly, were not detected in the paired healthy tissue from the same patient. In order to address this systematically, we combined information on variants from several different resources. However, filtering out all positions that vary in the reference genomes would affect true mutations and germline risk variants. In order to keep these, we excluded only hotspots that overlapped with the genome variants that occurred in more than 1% of the population. This does not exclude all false positives from the set, but it increases the quality of the current predictions.

To obtain the information on the common variants, we downloaded Ensembl variation data using the BioMart tool (Ensembl releases 75 and updated 82; the latter coordinates were transferred with the LiftOver tool to the genome assembly build 37). This included variation data from the 1,000 genomes (Genomes Project C *et al*, 2015), dbSNP (Sherry *et al*, 2001), and other databases. Furthermore, we downloaded the Kaviar database (Glusman *et al*, 2011) as well as coordinates of genomic variants collected by the Exome Aggregation Consortium (release 1, a subset without the TCGA variants) (Lek *et al*, 2016) and assessed overlaps with the identified hotspot mutations. We excluded all hotspots that overlapped with a variant for which the associated general population frequency in either of the reference populations was higher than 1%.

Next, we used the PolyPhen-2 tool (Adzhubei *et al*, 2010) through the online batch submission and retrieved predictions for the likely functional effects of the genomic changes corresponding to protein hotspots. For all coordinates that were predicted to result in a "deleterious" change, we kept the associated protein hotspot position. All other predicted hotspots were excluded from the further analysis in order to reduce possible false-positive instances of hotspot mutations in the final set. This, however,

possibly also excluded some true positives. In addition, we submitted the genomic coordinates of hotspots to the Provean tool (Choi *et al*, 2012) and noted that positions, which were predicted as "deleterious" according to this predictor largely agreed with the used predictions.

To obtain estimates of random mutation events, we made a simulation where amino acids were equally represented and all nucleotides were equally likely to mutate. We repeated the simulation 10,000 times and noted frequency of different types of events. This gave an estimate for the expected mutation rate of glutamate to lysine in a randomly composed and freely mutated protein set.

### Characterization of genes with hotspot mutations

We retrieved the list of genes annotated as the Cancer Gene Census (Futreal *et al*, 2004) from the COSMIC web resource (Forbes *et al*, 2017). We used the list available on January 11, 2017. Based on this catalogue, we classified the genes with the identified hotspot mutations as known or candidate cancer driver genes. Next, through the Ensembl BioMart service (Kinsella *et al*, 2011), we retrieved predictions for all human genes that are homologous to those with hotspot mutations, and assessed which of these were in the cancer Gene Census or had a hotspot mutation themselves.

We composed two other sets of genes to which we compared those with hotspot mutations: (i) all other genes in the Cancer Gene Census, that is, excluding those with hotspots and paralogs of genes with hotspot mutations (based on the Ensembl homology assignments) and (ii) all other protein coding genes in the UniProtKB reference set of human proteins (release 2016_06, July 2016), that is, excluding all Cancer Gene Census genes or genes with hotspot mutations. We used the UniProtKB and Ensembl BioMart to map Ensembl, Uniprot, and Entrez gene identifiers for the genes in the three sets.

Next, we obtained information on the proteins classified as enzymes from the Expasy ENZYME nomenclature database (Bairoch, 2000), release 06-July 2016. We compared the fractions of enzymes in the three datasets using the chi-squared test in R. Further, we retrieved GO annotations (Gene Ontology, 2015) for the genes in the three sets using the Ensembl Biomart service. In addition, we obtained information on the domains encoded by the proteins in the three sets by taking predictions for the human proteins provided by the Pfam database (release 28.0, May 2015). For the GO terms and Pfam domains associated with the set of genes with hotspot mutations (i.e., with three or more genes), we compared fractions of proteins annotated with the same terms and domains among the three sets. For this, we used the Fisher's test in R and accounted for the multiple testing with the Benjamini–Hochberg correction (https://www.r-project.org/).

Genes involved in the signaling by Rho GTPases were obtained from the GenScript website where the assignments were based on the Reactome annotations (Fabregat *et al*, 2016). Rho proteins often function as binary switches that control a variety of biological processes (Ellenbroek & Collard, 2007).

### Annotation of the affected protein residues

We used the Pfam batch service (Finn *et al*, 2016) to predict exact domain coordinates in the representative Ensembl protein sequences, which the cancer mutations were mapped to. We

overlapped positions of hotspot mutations with the boundaries of individual domains. In addition, we retrieved data on functionally annotated regions within the proteins with hotspot mutations from the UniProtKB service. We aligned these protein regions to the representative Ensembl proteins and looked for an overlap with the hotspot mutations. UniProtKB annotations also provide information on the PDB structures associated with each protein. For all proteins with hotspot mutations, for which X-ray structural data were available, we obtained a representative PDB structure. Representative structure was chosen so to cover the longest possible stretch of a protein sequence and to have a resolution below 3 Angstroms (Å), when possible. In addition, for the candidate proteins that were not in the Cancer Gene Census, we analyzed all available PDB crystal structures. For the selected PDB identifiers, we retrieved residue-associated data from the PDBePISA database (Krissinel & Henrick, 2007). Of note, from the structural point of view, residues are classified as being either at the interface, buried inside the protein core or exposed at the surface (Chothia, 1976; Levy, 2010). The PDBePISA assignments provided information on the accessible and buried surface area of each amino acid. In our analysis, we focused on the PDBePISA annotated complex interfaces that had a complexation significance score (CSS) higher than 0.3 when the interface contacts were formed between two proteins, or CSS higher than 0.05 when the interface was formed with a ligand or nucleic acid. The score indicates how significant the interface is for the assembly formation, and we used these thresholds to prioritize biologically relevant interfaces for further analyses. We further classified protein residues with the buried surface area larger than 25 $Å^2$ as interface residues. We used a conservative threshold in order to obtain a more confident prediction of amino acids that contribute to the interface formation. Interfaces encompassed protein–protein, protein–DNA, and protein–ligand interactions. We mapped coordinates of interfaces in the PDB structures to Ensembl proteins by aligning interface stretches to these and, when interface segments were shorter than five amino acids, we included surrounding residues in the alignment so to have segments of five amino acids or longer. To align the PDB with the Ensembl protein sequences, we used a Perl script and a Smith–Waterman local alignment algorithm from the Emboss package with the default parameters (release Emboss-6.6.0). Coordinates were mapped when the sequence identity between the aligned peptides was at least 80%, or—if the alignment was longer than ten amino acids—at least 60%. We then assessed an overlap of hotspot positions with the interaction interfaces in proteins.

### Signatures of tumor suppressor genes

Somatic mutation files obtained from the TCGA and ICGC and described above also contained information on the other types of small-range mutations in the sequenced genomes. We analyzed these changes and classified premature nonsense mutations and out of frame insertions and deletions in the encoded proteins as highly deleterious. We compared these to the synonymous amino acid changes in the same proteins, which represented an imperfect (Supek *et al*, 2014) approximation for the local background mutation rate. Genes with a high number of the reported deleterious changes (at least 25) and large contribution of these changes to the overall mutation load (the ratio of deleterious over synonymous changes was higher than 0.7) were considered as tumor suppressor candidates.

To address whether a mechanism of action for any of the mutations was interference with the activation of tumor suppressors through phosphorylation, we obtained positions of phosphosites in these proteins using the PhosphoSitePlus resource (Hornbeck *et al*, 2012). We obtained coordinates of phosphosites that were reported by a small scale or two or more mass spectrometry studies and assessed if they were in the vicinity of hotspots.

### Interaction partners and neighborhoods of the proteins with hotspots

To compare a fraction of domains that mediate protein interactions among the previously composed sets of genes (genes with hotspot mutations, other genes in the Cancer Gene Census and background human proteins), we obtained domain annotations from the *i*Pfam 1.0 (June 2013) and used chi-squared test in R. Further, we used a Functional Annotation service from the ConsensusPathDB-human database compendium (Kamburov *et al*, 2013). This resource integrates human interaction networks and includes binary, complex, signaling, and other experimentally identified interaction relationships. Specifically, for the uploaded proteins with hotspot mutations, we used the Network neighborhood-based entity set analysis, which finds network centers whose direct interaction neighborhood is enriched with the provided proteins. Overrepresentation is calculated based on the background of all human proteins for which the annotation necessary for the analysis is available. The calculated *P*-value is based on the hypergeometric test and corrected for multiple testing with the false discovery rate method. Among the significant hits, we searched for the protein centers that had a hotspot mutation themselves and for which other proteins with hotspots were overrepresented in their interaction neighborhood.

In addition, we used publicly available annotated physical interactions from the BioGRID database (BioGRID version 3.4.138), and among these, we searched for the interaction partners of proteins with hotspot mutations. We considered only interactions classified as "physical association" or "direct interaction", and this included interaction pairs identified using the affinity chromatography or the BioID method for the former, or using the two hybrid, pull down, enzymatic studies, florescence energy transfer, Western blotting, or protein complementation assays for the latter. We then calculated the fraction of proteins from the Cancer Gene Census in the interaction neighborhood of each individual protein. Enrichment of known cancer drivers among the interaction neighbors of candidate proteins could serve as an indication of their connection to the cancer-relevant pathways.

### Clustering of tumor types

We grouped the TCGA and ICGC tumor types that had the same name abbreviation and excluded from the further analysis those, which had sequencing data available for fewer than 50 patients. In all other tumor types, we noted a number of patients with the somatic mutation data. For each of the identified hotspot mutations, we looked at the individual tumor types and calculated a fraction of patients which had this residue mutated. On the thus obtained table, we used the pvclust R package, which performs hierarchical clustering with bootstrap resampling (Suzuki & Shimodaira, 2006). We used median distance as a clustering method and noted the resulting clusters of tissue types that had high *P*-values after 10,000 bootstrapping repetitions.

We next drew a heatmap plot where the clustering of tissue types was defined with the pvclust results and where, for the clarity, we reduced the number of the visualized hotspots by including only those mutated in one-third or more of the patients in at least one tumor type. Clustering of hotspots was based on the pvclust results. The original values for the fractions of patients with each hotspot mutation were log2-transformed, and zeroes were represented as the lowest values in the table. A heatmap in Fig 5 depicts a simplified version of the pvclust dendrogram. R packages gplots and RColorBrewer were used for the heatmap drawing.

### Interface mutation clusters

We obtained a representative dataset of stable human protein interactors with a structural support from the Interactome3D database version 2016_06 (Mosca *et al*, 2013). The Interactome3D database contains experimentally observed protein interaction partners for which either a PDB structure exists or a homology model could be generated. In our analyses, we focused on those interaction pairs that are likely to form stable complexes [i.e., interactions that were reported within the same complex in the Reactome database (Fabregat *et al*, 2016) or Corum (Ruepp *et al*, 2010), or by three or more Affinity purification experiments (Chatr-Aryamontri *et al*, 2015)]. We obtained a representative dataset provided by the Interactome3D that corresponds to the top ranking structures and models. For all the template PDB structures corresponding to members of the above-defined complexes, we retrieved residue annotations from the PDBePISA database (Krissinel & Henrick, 2007). We further followed the procedure described above in the Methods section "Annotation of the affected protein residues" in order to identify interface residues in the template proteins and, when possible, map the obtained interface coordinates to the Ensemble protein sequences. Next, we used the ActiveDriver tool (version 0.0.10) (Reimand & Bader, 2013) in order to assess positional clustering of mutations reported in the cancer genomics studies. ActiveDriver implements a logistic regression model. It was initially developed to identify phosphosites that had a mutation pattern which significantly differed from the gene background mutation rate, while accounting for the differences in mutation rates between structured and disordered protein regions. We used the package settings in which the user can provide coordinates of the regions of interest and analyzed mutations clustering at the interaction interfaces. For the required mutation data, we provided the amino acid changes reported in the above-described TCGA and ICGC datasets, and we specified intrinsically disordered protein regions by using the IUPred tool (Dosztanyi *et al*, 2005; version 1.0) with the settings for short disorder and a residue disorder prediction threshold of 0.5. From the ActiveDriver predicted proteins with significant interface mutation clusters, we analyzed further those that had a multiple testing corrected *P*-value lower than 0.05. For these proteins, we searched for the particular interfaces that were likely to be affected with the reported mutations. We used GO term protein annotations (Harris *et al*, 2004; GO database annotations for UniProtKB proteins, release November 2017) to assess cellular processes in which these proteins could play a role, and the ConsensusPathDB service for finding larger affected complexes. For the former analyses, we compared the GO term frequency between the set of proteins with frequently mutated interfaces and the rest of the representative proteome using the Fisher's test and we adjusted the obtained *P*-values for multiple testing using the Benjamini–Hochberg correction. In the Dataset EV10, we reported significant terms which were associated with three or more proteins.

### Drug targets

We obtained information on the druggable proteins for the list of genes with hotspot mutations using the DGIdb service (Griffith *et al*, 2013) (http://dgidb.genome.wustl.edu/).

### Amplified and overexpressed genes

We used the cgdsr R package from the cBioPortal and obtained the GISTIC values and the estimated z-scores for mRNA values from the available TCGA studies, which included data from 35 and 32 tumor types, respectively. Using as thresholds CNV values > 1 and mRNA z-scores > 2, we looked at the percentages of patients in each tumor type that had the "hotspot genes" amplified and/or overexpressed. To obtain comparable sizes of gene sets, we asked for a gene to be either amplified in 5% or more or to be overexpressed in 10% or more of the patients of the same tumor type. We excluded ovarian adenocarcinoma data from the former analysis as the majority of genes had apparently unspecific and strong signals in this tumor type.

### Data availability

The code for identifying and annotating protein hotspot residues is available for download as an R/Bioconductor software package "*DominoEffect*" in Code EV1. The folder contains example datasets and a vignette that explains the code usage in detail. We recommend using the up-to-date version available from the Bioconductor website: https://bioconductor.org/packages/3.7/bioc/html/Domino Effect.html.

Expanded View for this article is available online.

### Acknowledgements

We gratefully acknowledge contributions from the TCGA Research Network. We are also very grateful to the ICGC for the tumor genome resequencing data generated. We would like to acknowledge Madan Babu, Jelena Bara-novic, Robert Weatheritt, Rodolfo Ciuffa, Steffie Revia, Marco Faini, Max Frank, Isabell Bludau, and Lucija Silic for their feedback on the study. We apologize for not including citations to individual cancer genomics studies due to space restrictions. A structure used in the scheme on Fig 1 is reprinted by permission from: Springer Nature, Nature Structural and Mole-cular Biology, The structure of the Pan2–Pan3 core complex reveals cross-talk between deadenylase and pseudokinase, IB Schafer *et al*, 2014 (Licence Number: 4303640958191). Work in the laboratory of M. Boutros is in part supported by an ERC Advanced Grant Syngene (ERC-2011-ADG_20110310). R.A. acknowledges the following grant support: ERC grant Proteomics v3.0 (ERC-2008-AdG_20080422), ERC Proteomics 4D (670821), and the Swiss National Science Foundation (3100A0-688 107679). M. Buljan was funded by the Cluster of Excellence CellNetworks (EXC 81) and SystemsX.ch (TPdF 2013/135) fellowships. P.B. was supported by the SNSF SystemsX.ch fellow-ship (TPdF 2013/134).

## Author contributions

MBu, PB, RA and MBo jointly conceived the study, interpreted the results, and wrote the manuscript. Analyses were performed by MBu. PB co-authored the *DominoEffect* R/Bioconductor package. RA and MBo co-supervised the study.

## Conflict of interest

The authors declare that they have no conflict of interest.

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
