## [Review Process File · Molecular Systems Biology]

Systematic characterization of pan-cancer mutation clusters

Marija Buljan, Peter Blattmann, Ruedi Aebersold and Michael Boutros

Review timeline:

Submission date:	3 September 2017
Editorial Decision:	6 November 2017
Revision received:	10 January 2018
Editorial Decision:	5 February 2018
Revision received:	14 February 2018
Accepted:	15 February 2018

Editor: Maria Polychronidou

Transaction Report:

1st Editorial Decision

6 November 2017

Thank you again for submitting your work to Molecular Systems Biology. We have now heard back from the two referees who agreed to evaluate your study. As you will see below, the reviewers are overall quite positive and think that the approach and findings seem interesting. They raise however a series of concerns, which we would ask you to address in a revision of the manuscript.

The reviewers' recommendations are rather clear and therefore I think that there is no need to repeat all the points listed below. Please let me know in case you would like to discuss further any of the issues raised by the reviewers.

REVIEWER REPORTS

Reviewer #1:

Buljan et al., in their manuscript "Systematic characterization of pan-cancer mutation clusters" perform a systematic analysis of drivers by investigating properties associated a protein's amino acid residue potential to drive tumorigenesis, by identifying hotspot regions within candidate protein-coding genes correcting for gene length, background mutation rates and presence of common polymorphisms. They report 180 such hotspot mutations, identifying over a hundred genes as potential drivers which not currently are in any Cancer Census list. The authors further demonstrate enrichment of these mutations among proteins that can exist in the on and off state and among conserved binding interfaces. These mutations often associate with tumor type. Their analysis reveals novel cancer candidate genes and further suggests an important role for the disruption of protein interactions in cancer. I found the manuscript to be overall well-written and logically structured, the advances properly documented, and its findings to be interesting and relevant - though I did identify a number of points that the authors will need to address to make their paper publishable in MSB.

- The introduction should clarify that approaches for driver mutation identification discussed here pertain only to coding regions (i.e. such as accessible by exome sequencing). Recent studies have made marked advances in studying point mutations, gene fusion events and somatic DNA rearrangements in intergenic regions with effects on cancer. These findings were not covered in the introduction - but should either be mentioned or the introduction should clarify that only mutations affecting protein-coding genes are at the focus.

- To account for sequence polymorphisms, the new tool the authors present - DominoEffect - excludes all genomic variants with a reported population frequency higher than 1% based on the 1000 Genomes Project, Kaviar and "other Ensembl-linked" databases. It was not clear to me whether these databases include ExAC, which currently, at the level of exomes provides the most comprehensive annotation of sequence polymorphisms (it contains data from >50,000 non-cancer patient individuals). If not, the authors will need to verify whether any of their candidate hotspot sites is marked as a sequence polymorphism (>1%) amongst relevant human populations in ExAC.

- A majority (106/160) of the genes with the here identified mutation hotspots were as yet not classified as known cancer drivers, though the authors report that several of these candidate genes (i.e. 18) had a homologue in the Cancer Gene Census. Does this reflect a statistical enrichment of such homologues amongst the 106 not yet classified candidate drivers?

- The authors provided comparisons to databases of cancer point mutations, but did not consider other types of DNA alterations known to be common (and highly relevant) in cancer. The authors should assess whether there is any enrichment of these 106 candidate drivers amongst those genes seen recurrently amplified or deleted in cancer genomes (e.g. according to GISTIC - Zack et al. Nat Gene 2013), or such activated by intergenic DNA rearrangements (e.g. Weischenfeldt et al. Nat Genet 2016).

- To my taste the authors overly use the term novel, which does not reflect good writing style. They should omit that term whenever possible -- explain the novelty in words without using the terms 'novel'. Or 'new'.

Reviewer #2:

Buljan and coauthors present a very interesting proteo-genomics study to characterize more than a million single nucleotide variant in thousands of cancer genomes to discover and characterize hotspot mutations. Functional interpretation of mutations is an active research field and incredibly important as we are sequencing more and more genomes to map variation and hunt genetic disease mechanisms. Analyses like this can reveal new mechanistic details about known cancer driver mutations and also reveal previously uncharacterized ones. The authors should be acknowledged for making their software available to the community. Admittedly this is not an entirely novel approach and several studies and software tools are present in the literature. There is a recent review paper on mutation analysis at a sub-gene level in cancer genomes, including algorithms that study hotspots mutations that looks relevant to the study [<https://www.nature.com/nmeth/journal/v14/n8/full/nmeth.4364.html>].

I am generally very positive about this manuscript however have some concerns regarding the computational model, analysis power and some conclusions. These should be addressed in detail prior to publication.

1. The statistical model has some ad-hoc approaches and parameters that raise concerns, although clarity is increased after reading the supplemental note (which I recommend to be shortened and added to the main methods). The authors have selected a window of 200 or 300 amino acids within the protein of interest to determine the significance of hotspot mutations, and further require that 15% mutations need to be in the same amino acid residue. This sets a number of constraints to statistical modeling. Couple of further points:

a - the median protein is probably around 4-500 amino acids in sequence length. would the model still work if the window-based constraint was removed and instead the entire protein sequence was counted as background?

b - how would the model perform on randomized data, e.g. when substitutions within a protein are randomly re-assigned to alternative amino acid positions using sampling with replacement? I would look into outliers like TP53 (many mutations, protein of average length) and TTN (many mutations, very long protein).

c - mutations per sequence position may be modelled better with Poisson rather than binomial statistics.

d - are the novel findings of hotspot-related genes usually longer genes? That would perhaps tell us something about the model.

2. As far as I understand, the entire analysis concerns a pan-cancer cohort of ~10,000 samples and 40 cancer types. Do the authors see cancer type specific hotspots in type-specific samples? Is there a correlation with cohort size or a lower bound of samples where the approach is not powered to detect hotspots?

3. Many numerical statements in the paper would benefit from statistical calculations of observed vs expected values. Not all analyses to be covered systematically, but some would definitely make the discussion of results more convincing. For example:

a - charged to polar amino acid (p4), charged to charged (p4) and several more in this paragraph

b - known cancer driver genes discovered, proportion of patients covered (top of p5)

c - kinase and RAS domains (p6)

d - .. others exist in the text

4. On/Off states of proteins is an interesting approach and concept; however it deserves more introduction as it may appear non-intuitive. I would recommend the authors seek orthogonal evidence to support this analysis, or alternatively de-emphasize and use more careful wording to better convey the idea. It seems that currently the authors just interpret lists of proteins with enzymatic function that are broadly known to have ON/OFF states. They do not really say whether a given mutation X brings a protein from ON to OFF or vice versa. This can be done using the resources of TCGA - for example a mutation hotspot in a transcription factor may change the expression of target genes and that could be a functional readout from matching transcriptomic data. Similarly, kinase hotspot mutations in activation loops may be apparent in downstream signaling cascades in RPPA data. I don't insist this is required for publication but would make the analysis and conclusions much stronger.

5. Hotspots inactivating tumor suppressors is also an interesting idea and deserves more discussion of potential mechanisms. Perhaps the hotspots affect post-translational modifications or short linear motifs known to activate tumor suppressors? Are they structurally important residues that change 3d confirmations?

6. TP53 is a major outlier in mutational analyses. How many of the mutations throughout the paper, in particular the clustering analysis on P9 are driven by any single gene such as TP53?

7. The use of the DAVID software for pathway enrichment analysis is potentially very problematic - <https://www.nature.com/nmeth/journal/v13/n9/full/nmeth.3963.html>.

Minor:

1. The manuscript would benefit from copy-editing. Some sentences can be simplified and wording rephrased.

2. one example - P3 "[DominoEffect] is insensitive to gene length, background mutation rates .. ". Perhaps the authors mean "robust" instead of "insensitive"?

3. hotspot is the major concept of this paper. It is worth defining it early on (amino acid? nucleotide? single or multiple residues) as the terminology in the community is not always uniform.

Reviewer #1:

Buljan et al., in their manuscript "Systematic characterization of pan-cancer mutation clusters" perform a systematic analysis of drivers by investigating properties associated a protein's amino acid residue potential to drive tumorigenesis, by identifying hotspot regions within candidate protein-coding genes correcting for gene length, background mutation rates and presence of common polymorphisms. They report 180 such hotspot mutations, identifying over a hundred genes as potential drivers which not currently are in any Cancer Census list. The authors further demonstrate enrichment of these mutations among proteins that can exist in the on and off state and among conserved binding interfaces. These mutations often associate with tumor type. Their analysis reveals novel cancer candidate genes and further suggests an important role for the disruption of protein interactions in cancer. I found the manuscript to be overall well-written and logically structured, the advances properly documented, and its findings to be interesting and relevant - though I did identify a number of points that the authors will need to address to make their paper publishable in MSB.

We thank the reviewer for the thoughtful and in-depth review of our manuscript and we also thank them for acknowledging that the presented findings are interesting and relevant for the field. We found their constructive criticism very useful and it helped us to further improve the manuscript. We address the comments in detail below. The changed text in the manuscript is marked in red.

1. The introduction should clarify that approaches for driver mutation identification discussed here pertain only to coding regions (i.e. such accessibly by exome sequencing). Recent studies have made marked advances in studying point mutations, gene fusion events and somatic DNA rearrangements in intergenic regions with effects on cancer. These findings were not covered in the introduction - but should either be mentioned or the introduction should clarify that only mutations affecting protein-coding genes are at the focus.

Thank you for pointing this out. We have added the following sentences to the introduction to state explicitly that we studied single amino acid changes, but also to acknowledge that this is only one of several mechanisms for introducing oncogenic changes:

(P2, L24-27): These mechanisms include genomic rearrangements, smaller insertions and deletions, or single point mutations. Eventually, they all lead to changes in the expression levels or to altered functions of cancer driver genes and their products.

(P3, L18-20): ... single protein residues that accumulated point mutations at a significantly higher rate than their surrounding sequence. We use the term 'Hotspot' to refer to such protein residues.

2. To account for sequence polymorphisms, the new tool the authors present - DominoEffect - excludes all genomic variants with a reported population frequency higher than 1% based on the 1000 Genomes Project, Kaviar and "other Ensembl-linked" databases. It was not clear to me whether these

databases include ExAC, which currently, at the level of exomes provides the most comprehensive annotation of sequence polymorphisms (it contains data from >50,000 non-cancer patient individuals). If not, the authors will need to verify whether any of their candidate hotspot sites is marked as a sequence polymorphism (>1%) amongst relevant human populations in ExAC.

Many thanks for highlighting this issue. It was not explicitly addressed in the original submission even though we did check for the overlap when the ExAC was published. We now obtained the new release of the ExAC dataset (a subset without the TCGA variants) and we refer to this analysis in the Methods (P16, L21-23) and Results (P4, L26). None of the ExAC variants that corresponded to the here-identified hotspots was reported with a frequency of 1% or higher in the sequenced cohorts. Therefore, our final list of the hotspot residues did not change.

3. A majority (106/160) of the genes with the here identified mutation hotspots were as yet not classified as known cancer drivers, though the authors report that several of these candidate genes (i.e. 18) had a homologue in the Cancer Gene Census. Does this reflect a statistical enrichment of such homologues amongst the 106 not yet classified candidate drivers?

Thank you for raising this point. We now compared the fraction of Cancer Gene Census homologs in the set of the ‘candidate’ genes with hotspots (i.e. 18/106) to the background of human representative proteins excluding the known cancer drivers. We found that the difference was significant and we added the following sentence to the manuscript (P5, L22-25):

The fraction of Cancer Gene Census homologs in the background set of human genes was significantly lower than among the hotspot genes (7 versus 17%, $p < 3 \times 10^{-4}$, Chi-square test).

4. The authors provided comparisons to databases of cancer point mutations, but did not consider other types of DNA alterations known to be common (and highly relevant) in cancer. The authors should assess whether there is any enrichment of these 106 candidate drivers amongst those genes seen recurrently amplified or deleted in cancer genomes (e.g. according to GISTIC - Zack et al. Nat Gene 2013), or such activated by intergenic DNA rearrangements (e.g. Weischenfeldt et al. Nat Genet 2016).

A short summary: We performed the analyses suggested by the reviewer (details are below) and they indicated a minor overlap of the hotspot genes with the genes within genomic amplification and deletion peaks reported by Zack et al. (11 genes) and no overlap with the genes activated by intergenic DNA rearrangements reported by Weischenfeldt et al. We then used a gene set from another study that analyzed pan-cancer data with the aim to identify proteins with frequent point mutations (Lawrence et al. Nature 2014) and compared it to these two studies: Again, the overlap of this dataset with the Zack and Weischenfeldt gene sets was minor (i.e. 19 and 1, respectively, out of 293 genes). However, even though they comprised only few genes, both GISTIC overlaps were higher than expected by random. We now include in the Dataset EV5 hotspot genes that were within genomic deletion peaks.

Additionally, in order to address the question of how gene expression changes and hotspot mutations relate to each other, we performed an independent analysis and asked if any

of the ‘hotspot genes’ were often found amplified or overexpressed across different tumor types. We refer to these results in the manuscript (P12, L40-41) and report them in the Dataset EV11.

Comparison with GISTIC peaks (Zack et al. Nat Genet 2013): We used the genomic coordinates that were reported for amplification and deletion peaks in cancer by Zack et al. and we obtained the Ensembl IDs for the genes that mapped within these regions (using the release Feb 2014 that corresponds to the reported hg19 coordinates). For the subsequent analyses we selected peaks with fewer than 25 genes, as the authors did in the original study. This resulted in a gene set consisting of 300 and 268 genes present within amplification and deletion peaks, respectively. The overlap with the set of genes with hotspot mutations was rather small, but it was still higher than expected from the overlap with background human genes. In total 7 out of 54 hotspot genes that were in the Cancer Gene Census (CGC) were in the peaks (4 genes within the Amplification peaks - compared to an expected 0.5 from the background gene set, and 3 within Deletion peaks - compared to an expected 0.6). In addition, 4 out of the 106 ‘candidate hotspot’ genes were in the Deletion peaks (the expectation from the background set was to have 1.2). We added these genes to the Dataset EV5 that lists candidate tumor suppressors. A particularly interesting instance is the *AGBL4* carboxypeptidase that was in the deletion peak with only one other gene and, due to a high mutation rate, was also highlighted in the original Supplementary table by Zack et al.

To assess if the low number of genes observed in the overlap was unexpected, we used a set of genes reported by another pan-cancer study that focused on point mutations (Lawrence et al Nat 2014). This set of genes included in total 293 genes with the mapped identifiers. A fraction of the CGC genes reported by Lawrence et al. that was in the amplification and deletion peaks was comparable to what we observed for the CGC hotspot genes (16 out of 134 for Lawrence et al.). Similarly, the overlap between other genes in their set and GISTIC peaks was minor (1 gene in an amplification peak, 2 in deletion peaks, out of the total 159 ‘non-CGC’ genes).

In conclusion, there seems to be a low overlap between hotspot mutations and regions of copy-number variation loci, which is nevertheless higher than expected by chance. The degree of overlap between our study and Zack et al. is similar to the overlap obtained by comparing the GISTIC set with another gene set identified through frequent point mutations in protein-coding regions (Lawrence et al. 2014).

Intergenic rearrangements in cancer: The Weischenfeldt et al. study was focused on identifying genes that associate with changes in regulatory elements that influence gene expression levels. This could be a complementary mechanism to hotspot mutations that potentially modulate protein activity. We did not observe any overlap between the genes reported by Weischenfeldt et al. and the hotspot genes (the authors list 18 genes significant on the pan-cancer level and 99 on the individual tumor level). To assess if this observation was expected, we again used the Lawrence et al. set with pan-cancer point mutations. Here, we also found an overlap with only one of the genes in the Weischenfeldt pan-cancer set and one in the individual tumor set.

Hence, there seems to be generally a low overlap between the genes that are strongly affected by intergenic DNA rearrangements in cancer and those that tend to accumulate point mutations at a high rate.

Analysis of TCGA data for the amplified and over-expressed genes: Even though genes with hotspot mutations did not seem to be strongly associated with the peak regions of genomic amplification, we performed a more comprehensive analysis assessing if any of the ‘hotspot genes’ were still frequently found amplified or overexpressed in the patients. For

this, we used the `cgdsr` R package from the cBioPortal. We obtained the GISTIC values and the estimated z-scores for mRNA values from the available TCGA studies; this included data from 35 and 32 tumor types, respectively. Using as thresholds CNV values greater than one and mRNA z-scores greater than 2 we looked at the percentages of patients in each tumor type that had the 'hotspot genes' amplified and/or overexpressed. To obtain comparable sizes of gene sets, we asked for a gene to be either amplified in at least 5%, or to be overexpressed in at least 10% of the patients of the same tumor type. There were in total 37 and 33 genes, respectively, that passed this condition in five or more different tumor types. The majority, i.e. 24, of these genes was shared between the two datasets. The GISTIC set here also included 3 out of 4 genes found in the amplified genomic peak regions (by Zack et al.). Compared to other genes with hotspots, the 24 shared genes were enriched in the GO term 'localization at the apical plasma membrane (5 genes, 14 times enrichment, $p < 0.08$), and 10 of them had a GO term that indicated that they associated with the plasma membrane (i.e. 45% of the 22 genes for which GO annotations were available; For a comparison, 28% of other hotspot genes had the 'plasma membrane' GO term assigned to them). We now include an additional **Dataset EV11** that lists genes identified in this analysis and we refer to it in the text (P12, L40-41).

5. To my taste the authors overly use the term novel, which does not reflect good writing style. They should omit that term whenever possible -- explain the novelty in words without using the terms 'novel. Or 'new'.

We re-wrote all sentences that contained the terms 'new' or 'novel' and that referred to the 'hotspot genes' that were not in the Cancer Gene Census (changes are marked red in the text). Thank you for the comment.

Reviewer #2:

Buljan and coauthors present a very interesting proteo-genomics study to characterize more than a million single nucleotide variant in thousands of cancer genomes to discover and characterize hotspot mutations. Functional interpretation of mutations is an active research field and incredibly important as we are sequencing more and more genomes to map variation and hunt genetic disease mechanisms. Analyses like this can reveal new mechanistic details about known cancer driver mutations and also reveal previously uncharacterized ones. The authors should be acknowledged for making their software available to the community. Admittedly this is not an entirely novel approach and several studies and software tools are present in the literature. There is a recent review paper on mutation analysis at a sub-gene level in cancer genomes, including algorithms that study hotspots mutations that looks relevant to the study [<https://www.nature.com/nmeth/journal/v14/n8/full/nmeth.4364.html>].

I am generally very positive about this manuscript however have some concerns regarding the computational model, analysis power and some conclusions. These should be addressed in detail prior to publication.

We would like to thank the reviewer for the in-depth review of our work and very thoughtful comments that we address below in detail. We also thank the reviewer for acknowledging our work on understanding molecular principles that define driver mutations and in making the analysis tool available to the community in the form of an R package. Following reviewer's comments, we introduced several changes to the package and we believe this substantially improved the tool. The package and its vignette are attached as separate documents. We will submit it to Bioconductor upon the manuscript acceptance, as during the review process there the package is visible to the general public.

In addition to this, we performed several analyses to clarify concerns raised by the reviewer and we provide a detailed response to individual comments below. We also included the manuscript mentioned by the reviewer in the introduction (P2, L45) as it indeed provides a comprehensive overview of the field. The changed text in the manuscript is marked in red.

1. The statistical model has some ad-hoc approaches and parameters that raise concerns, although clarity is increased after reading the supplemental note (which I recommend to be shortened and added to the main methods). The authors have selected a window of 200 or 300 amino acids within the protein of interest to determine the significance of hotspot mutations, and further require that 15% mutations need to be in the same amino acid residue. This sets a number of constraints to statistical modeling.

Thank you for the very useful comments on the statistical aspects of the tool. As suggested, we have now moved the supplemental text to the main Methods (P14, L17-26). Furthermore, we have adapted the R package so that the user can easily select and change parameters for identifying hotspot residues and thus easily observe dependencies between the chosen parameter set and results (described in the vignette). We still put emphasis on individual residues that accumulate a high fraction of a mutation load rather than on identifying all protein residues that mutate at a higher than expected rate.

As mentioned by the reviewer, we used fixed parameters for the window size and percentage of mutations when obtaining the set of hotspot residues. We now explain in detail below and in a summarized form in the manuscript (P14, L19-26) the rationale for selecting these parameters. A window-based approach was essential for detecting hotspots in driver proteins that are longer than the average, whereas the used percentage threshold ensured that we have a high fraction of true positives in the set. However, following the Reviewer's comments we now introduce recommendations for modifying the thresholds and for applying alternative approaches in the analyses that aim at obtaining as broad a set of candidate residues as possible.

Couple of further points: (a-d)

a - the median protein is probably around 4-500 amino acids in sequence length. would the model still work if the window-based constraint was removed and instead the entire protein sequence was counted as background?

a) To address this question, we used the tool with the same settings as in the manuscript, i.e. a hotspot residue had to be mutated in 5 or more samples and at least 15% of mutations had to map to the same residue, but as a background we used mutations across the whole protein. This resulted in fewer detected proteins and 92 of the 160 analyzed hotspot genes were not identified. The list of genes that were not detected with these parameters included almost half (i.e. 25 out of 54) of the hotspot genes that were in the Cancer Gene Census, in

particular those encoding longer proteins (e.g. ERBB2 – 1255 aa, EGFR – 1210 aa, KIT – 976 aa). We then tried the same analysis requiring only 10% of the mutations to map to the same residue, but again did not identify known hotspots in 15 cancer drivers that had an above average protein length. Hence, it seems that using the whole protein length as a background could considerably reduce the power of our analysis and introduce a bias towards drivers of shorter sequence length. We believe this makes a case for considering a windows-based approach in the analysis of hotspots (even though the whole protein approach can be useful if there are several hotspots in the same window).

We have changed the software to now allow for the user to easily modify the window length and compare results. As a default value we would still recommend to use 200 amino acids (aa), which should include an independent functional unit within a protein, and to report only hotspots that also pass the threshold on another window (we used the window of 300 aa). Of note, we have also tested a window of 100 aa: this reported twice as many hotspot genes as the 300 aa window, but a percentage of genes in the set that were associated with KEGG “pathways in cancer” decreased from 10 to 6%. In addition, in the revised software we now provide an option to use not only percentage (that should be adapted to a window length), but also overrepresentation, which is calculated as a number of observed over the number of expected mutations. This allows filtering based on both the Poisson p-value and the effect size (P 15, L 28-30, and the package vignette and code).

b - how would the model perform on randomized data, e.g. when substitutions within a protein are randomly re-assigned to alternative amino acid positions using sampling with replacement? I would look into outliers like TP53 (many mutations, protein of average length) and TTN (many mutations, very long protein).

b) We used the following three genes to assess how the tool performs on the randomized mutation data: (i) GNA11 protein with a typical hotspot mutation, (ii) TP53 that accumulates mutations with a high rate over the whole sequence, but also has ‘hotspot residues’ that inactivate the protein in a dominant negative manner and are hence described as “oncogenic mutations” (*Carol L Prives: The two faces of p53: Tumor suppressor and oncogene, 2012*) and (iii) TTN that has a high mutation rate over the whole sequence. Diagrams of pan-cancer mutations within these proteins are shown below. We counted the total number of mutations per each protein, randomly assigned the observed mutations across the protein length and ran *DominoEffect* with the settings used in the manuscript. We repeated this 100 times and did not observe a significant result even once. We then decreased the minimum number of mutations that had to occur at the same position to 3 or more and repeated the same analysis – this resulted in only one hit for TTN. This demonstrates that in the instances where background mutations occur with equal probability across different positions in the same protein, the model should perform without significant false positives. In our view, most false positive assignments are likely to come from the common polymorphisms that are as yet not annotated.

c - mutations per sequence position may be modelled better with Poisson rather than binomial statistics.

Thank you for this suggestion. We have now replaced the binomial with the Poisson statistics. (P 15, L 31, and the package vignette and code).

d - are the novel findings of hotspot-related genes usually longer genes? That would perhaps tell us something about the model.

d) We compared lengths of representative proteins for the here-identified Hotspot genes – separately for those in the Cancer Gene Census (CGC) and others, for the other CGC genes and for all other human genes (Background genes). We found that a distribution of protein lengths among the genes with hotspots was in general closer to the CGC genes than to the background genes. Medians of protein lengths were:

Hotspot genes: 686 aa (hotspot CGC genes) and 590 aa (Other hotspot genes)
 Other CGC genes: 635 aa and
 Other Background genes: 434 aa.

There were no statistically significant differences in the length distributions among the first three sets, but these all differed from the background human genes ($p < 5 \times 10^{-4}$). Distribution of protein lengths is shown below (cut to 4,000 aa, hotspot genes that are in the CGC are colored red and other are colored blue). In conclusion, genes with hotspot residues, which are not in the Cancer Gene Census, are in general not longer than known cancer driver genes.

2. As far as I understand, the entire analysis concerns a pan-cancer cohort of ~10,000 samples and 40 cancer types. Do the authors see cancer type specific hotspots in type-specific samples? Is there a correlation with cohort size or a lower bound of samples where the approach is not powered to detect hotspots?

Cohort size

With the default analysis settings, a major limiting factor in our approach was imposed by the requirement that a mutation needs to be reported in 5 or more samples (assuming that the respective region does not accumulate background mutations at a high rate). Our main goal here was to systematically characterize properties of protein hotspot residues and for this we needed both a large set of such protein positions, but also a low false positive rate.

When the tool is used so to detect as many hotspots as possible then the limiting factors are: (i) a cohort size, (ii) a frequency of the hotspot mutation and (iii) the background mutation rate in the surrounding region, which will also partly depend on a tumor type. If one requires at least 5 mutations at the same residue, a naïve estimate is that for a hotspot residue that is mutated in 10% of the patients with a certain tumor type one would need more than 80 samples to identify the hotspot with the confidence interval of 80% and a margin of error of 4% (if we use the formula: $n = (z/M)^2 \times p(1-p)$, where M is the margin error and z is the z -value for the determined confidence interval). Below, we discuss results of the analysis of hotspots in the individual tumor types: the smallest cohorts in which we detected hotspots were Diffuse Large B-cell Lymphoma and Uterine Carcinosarcoma with 48 and 56 samples, respectively. In these cases the detected hotspots were present in 10% or more of the sequenced samples. Among ovarian cancer patients, where the cohort size was close to 500 samples, it was possible to identify KRAS hotspot, which was mutated in 1% of the samples.

Recent simulations that estimated the power of the MutSig tool (Lawrence et al. Nature 2014, Fig 5) have also shown that with the current cohort sizes genes mutated at 5% or less above background cannot be reliably detected for most tumor types. We now refer to this in the discussion and methods (P13, L22-24; P14, L36-38) and in the package vignette we give guidelines for relaxing thresholds when exploring cancer type-specific hotspots.

Analysis of hotspots in the individual tumor types

To assess if there are tumor type-specific mutations that were not detected on the pan-cancer level, we used *DominoEffect* package with the default settings and analyzed individual TCGA tumor datasets (in total 29 different types, COAD and READ merged together). This highlighted only a handful of additional genes. Notably, the above-mentioned hotspots in TP53 were detected in seven different tumor types. Biologically relevant examples also include a hotspot within the FOXA1 gene in breast cancer, RHOA in stomach cancer and CRIPAC in adrenal carcinoma. With this analysis in regard, we also include a comment in the paragraph on the clustering of tumor types (P10, L40-42).

Additional notes on this analysis: We identified hotspot mutations in all but three tumor types (CHOL with 35 samples, KICH with 66 and SARC with 247). Two datasets, however, had an apparently high rate of false positive hotspots: colorectal and skin cutaneous melanoma (SKCM). The former dataset included a very high fraction of common variants, and a possible explanation for the latter one is that the SKCM samples have a high mutation rate due to UV and there is a strong bias with regard to how likely each genomic position is to mutate. This has been discussed previously (Supek and Lehner: Clustered mutation signatures reveal that error-prone DNA repair targets mutations to active genes (Cell, 2017)).

3. Many numerical statements in the paper would benefit from statistical calculations of observed vs expected values. Not all analyses to be covered systematically, but some would definitely make the discussion of results more convincing. For example:

Many thanks for pointing to these. We calculated the respective expected values and added these to the manuscript (pages and lines are indicated below):

a - charged to polar amino acid (p4), charged to charged (p4) and several more in this paragraph

a – We estimated the frequency of random events using simulated mutations in random positions in the codons. We changed the corresponding text in the Results (P5 L4-8) and in the Methods (P16, L35-39).

b - known cancer driver genes discovered, proportion of patients covered (top of p5)

b – We compared the number of cancer drivers in this set to a random set of genes that accumulated mutations at a high rate (i.e, more than 100 in total within a gene) or that had an individual residue with 5 or more mutations. We added the observed overrepresentations to the text (P4, L42 - P 5, L1). We also composed a random sample of 180 positions that had 5 or more mutations (with 1,000 repetitions) and assessed a fraction of patients that had at least one of the residues mutated. The analyses showed enrichment of the here-identified hotspot genes in the CGC genes and a higher than expected fraction of patients with a mutation in at least one hotspot residue (P5, L12-15).

c - kinase and RAS domains (p6)

c - We used a conservative approach and looked only at the sequences of proteins containing kinase and Ras domains. We found that for both the Kinase and Ras proteins there was a tendency for mutations to map within the respective domains. We added a comment to the text (P7, L7-10). For the proteins with Ras and its homologous G-alpha domain the p-value

was less significant, but a majority of the mutations were mapping within only few amino acids responsible for binding GTP.

d - .. others exist in the text

d – We also added to the text the expected frequency of Cancer Gene Census homologs (reply to the Reviewer #1, **P5, L22-24**) and in several places we indicated an overrepresentation compared to the expectation based on the background frequency (changed text is always in red).

4. On/Off states of proteins is an interesting approach and concept; however it deserves more introduction as it may appear non-intuitive. I would recommend the authors seek orthogonal evidence to support this analysis, or alternatively de-emphasize and use more careful wording to better convey the idea. It seems that currently the authors just interpret lists of proteins with enzymatic function that are broadly known to have ON/OFF states. They do not really say whether a given mutation X brings a protein from ON to OFF or vice versa. This can be done using the resources of TCGA - for example a mutation hotspot in a transcription factor may change the expression of target genes and that could be a functional readout from matching transcriptomic data. Similarly, kinase hotspot mutations in activation loops may be apparent in downstream signaling cascades in RPPA data. I don't insist this is required for publication but would make the analysis and conclusions much stronger.

Thank you for the comment and for discussing this. We have reorganized the text in the manuscript and deemphasized the message (including a changed paragraph title, **P6, L44**). In the text, we highlight the findings that genes with hotspots are enriched in enzymes in general, but also specifically in protein classes that are explicitly described as ‘binary switches’ in the literature: Ras proteins and proteins involved in the signaling by the Rho GTPases (changes on: **P5, L35-39; P6 L1-2; P6, L21-24; P6, L39-41**; we additionally deleted the strong claims on the same pages). Several of the well-studied hotspots within kinase and GTPase domains are known to directly impact the “on/off switch” in these proteins and consequentially cause a cascade of downstream events. However, it is true that the instances with a clear functional link make only a small fraction of all cases. Since we do not have a direct evidence for the effects of the majority of mutations we toned down the text in the Results section.

RPPA analysis that we now performed but did not include in the manuscript:

Independently, we obtained the TCGA pan-cancer RPPA dataset (Li et al: TCPA: a resource for cancer functional proteomics data), but from the merged data it was not possible to identify unambiguous and conclusive trends even for the well known oncogenic hotspots. For instance, samples with the IDH1 mutation were significant in the majority of the assays (191 of 258), but a very large fraction of other hotspots did not significantly differ from the random samples we used for testing. Several of the observed instances were indeed supported by the literature (SMAD4 and SNAIL, POLE and PCNA, RXRA and SRC, TSPAN9 and AKT assay), but since the analysis overall was not conclusive we decided not to include this in the manuscript. Also, in general, when only a small set of patients has a certain mutation (for instance 10 or less, which is a great majority of the TCGA hotspots), it might be quite challenging to find unambiguous trends in the transcriptomics or RPPA data.

5. Hotspots inactivating tumor suppressors is also an interesting idea and deserves more discussion of potential mechanisms. Perhaps the hotspots affect post-translational modifications or short linear motifs known to activate tumor suppressors? Are they structurally important residues that change 3d conformations?

Because of the filtering step that prioritized conserved residues and which was a necessary step for avoiding non-annotated common polymorphisms, it is possible that we missed some of the hotspots within faster evolving phosphorylation sites. To further assess mechanisms of tumor suppressor inactivation among the detected hotspots, we searched the literature and we additionally obtained phosphosite annotations from the PhosphositePlus database (we focused on the residues reported by a small scale study or two or more mass spectrometry studies). Roughly, hotspots within tumor suppressors with available annotations could be classified as (i) those that inactivate the protein, and because the protein forms oligomers they act as dominant negative and also inactivate the other copy with the wild-type amino acid (Pten, Fbxw7, TP53), (ii) mutations that occur at the interface that regulates the oncogene (Pik3r1) or is necessary for forming an interaction that activates the tumor suppressor (Smad4) (iii) mutations that are at a residue essential for substrate binding (Crebbp or Ep300) or finally, as the phosphosite analysis now highlighted: (iv) mutations that can affect activation of the tumor suppressor by interfering with the phosphorylation (hotspot in the Chek2 kinase is within the activation loop and close to three phosphorylation sites; mutation in the Flt3 kinase also maps to an activation loop and EZH2 hotspot is a tyrosine that can be phosphorylated). We added these annotations and relevant literature citations to the Dataset EV5, and we now refer to the possible mechanisms of action in the Discussion, where we summarize these observations and highlight the Chek2 example (P12, L34-38). Of note, there were also three more tumor suppressors where the hotspot was close to a phosphosite, but we were not confident that this affected the mechanism of action so we did not add these to the EV5 (Dlc1, Fubp1, Idh2).

6. TP53 is a major outlier in mutational analyses. How many of the mutations throughout the paper, in particular the clustering analysis on P9 are driven by any single gene such as TP53?

As discussed above, with the parameters we used in this study, hotspot residues in the TP53 were not detected on the pan-cancer level and TP53 was not one of the 160 genes. However, clustering analysis is based on the percentages of patients with each mutation and the frequently mutated (and well-studied) hotspots contribute to forming the major clusters (27% of the analyzed samples had a mutation in at least one of only 8 genes, Fig 2A). We now assessed if also a single mutation was sufficient to distinguish major trends. KRAS-12 is mutated in 841 (i.e. 8%) of the total samples (TCGA + ICGC) and this mutation alone was sufficient to distinguish some of the adenoma samples (lung adenocarcinoma and colorectal adenocarcinoma, Figure below left). Adding the next two residues with the highest frequency in the pan-cancer dataset (BRAF-600 and IDH1-132) further distinguished tumor types in which these mutations were highly prevalent (thyroid and skin cancer for BRAF and low grade glioma for IDH1, not shown). Finally, clusters obtained using only information on the 8 most commonly mutated genes are shown on the right below: this alone separated several of the related tumor types, but adding information on more hotspots, as in the manuscript, was more powerful in defining the relationships. We added a sentence referring to this to the Discussion (P13, L19-22).

Hierarchical clustering dendrograms (complete linkage) for Kras alone and for the top 8 proteins:

7. The use of the DAVID software for pathway enrichment analysis is potentially very problematic –

<https://www.nature.com/nmeth/journal/v13/n9/full/nmeth.3963.html>.

There was indeed a long gap between the two latest updates of the DAVID database (6.7 in Jan 2010 to 6.8 in May 2016). We used the release 6.8 so we did not expect the consequences to be as dramatic as those discussed in the reference above. However, to be sure, we obtained the latest GO term annotations for the UniProt reference proteome and repeated the analysis using these annotations. The list of significant terms is now provided in the Dataset EV 10. GO terms are more detailed, but overall they are highly related to the cellular processes previously identified with DAVID (signal transduction, cell proliferation, etc.). We changed the text to refer to this analysis (P11, L21-22 and Methods P20, L19-23).

Minor

The manuscript would benefit from copy-editing. Some sentences can be simplified and wording rephrased.

- one example - P3 "[DominoEffect] is insensitive to gene length, background mutation rates ..". Perhaps the authors mean "robust" instead of "insensitive"?

Thank you for emphasizing this. We have copy-edited several parts in the manuscript (including the sentence with 'robust'). All changes in the manuscript are marked in red.

Hotspot is the major concept of this paper. It is worth defining it early on (amino acid? nucleotide? single or multiple residues) as the terminology in the community is not always uniform.

Many thanks for this comment. It is true that "hotspots" have been used in different contexts, also as genomic and 3D hotspots. To avoid any potential confusion, we have now clarified this in the introduction (P3, L18-20).

2nd Editorial Decision

5 February 2018

Thank you for sending us your revised manuscript. We have now heard back from the two referees who agreed to evaluate your study. As you will see below, the reviewers think that the study is now in principle suitable for publication.

Reviewer #2 recommends the inclusion of benchmarking analyses using simulated mutation data. While we do not think that these analyses are mandatory for the acceptance of the study, we would not be opposed to their inclusion in case you would decide to add them to the study.

Before we formally accept the manuscript for publication, we would like to ask you to address the following editorial issues listed below.

REVIEWER REPORTS

Reviewer #1:

The authors have done a great job in revising their manuscript.

Reviewer #2:

The authors have greatly improved their manuscript and I support its acceptance for publication in principle.

That said, I would recommend expanding their method benchmarking that makes use of simulated (shuffled) mutation data and adding it to the manuscript (in response to my comment 1b). More genes should be included in the analysis. Evaluating the method's performance on randomly simulated data is strongly recommended for bioinformatics methods and the presence of such an analysis would increase confidence of users of this software.

2nd Revision - authors' response

14 February 2018

We have included a sentence to the manuscript (second paragraph in the first section of the Results) to address the comment raised by the Reviewer 2 regarding the simulated mutation data.

Corresponding Author Name: Ruedi Aebersold and Michael Boutros

Manuscript Number: MSB-17-7974